# Amino acid insertion in Bat MHC-I enhances complex stability and augments peptide presentation
Suqiu Wang [1], Liangzhen Zheng[2,3], Xiaohui Wei[4], Zehui Qu [5], Liubao Du[1], Sheng Wang[3,6] & Nianzhi Zhang [1] ✉

Bats serve as reservoirs for numerous zoonotic viruses, yet they typically remain asymptomatic owing to their unique immune system. Of particular significance is the MHC-I in bats, which plays crucial role in anti-viral response and exhibits polymorphic amino acid (AA) insertions. This study demonstrated that both 5AA and 3AA insertions enhance the thermal stability of the bat MHC-I complex and enrich the diversity of bound peptides in terms of quantity and length distribution, by stabilizing the $3_{10}$ helix, a region prone to conformational changes during peptide loading. However, the mismatched insertion could diminish the stability of bat pMHC-I. We proposed that a suitable insertion may help bat MHC-I adapt to high body temperatures during flight while enhancing antiviral responses. Moreover, this site-specific insertions may represent a strategy of evolutionary adaptation of MHC-I molecules to fluctuations in body temperature, as similar insertions have been found in other lower vertebrates.

COVID-19 has resulted in immense global devastation and loss, with its origins traced back to a bat-borne coronavirus[1,2]. Bats harbor a multitude of viruses, including rabies[3], Hendra virus (HEV)[4,5], Nipah virus[6,7], severe acute respiratory syndrome coronavirus[8–10], Middle East respiratory syndrome coronavirus[11–13] and Ebola virus[14], all of which pose a threat to human and other mammals. Due to their diverse species composition[15,16], widespread distribution[17], and unique ability for sustained flight, bats are considered significant hosts of zoonotic viruses, contributing to the emergence of infectious diseases. Astonishingly, despite being carriers of numerous lethal viruses, bats do not display obvious pathological symptoms[18–21]. This remarkable resilience can be attributed to their exceptional immune system, which combines heightened host defense responses with immune tolerance, effectively combating viral infections while suppressing excessive inflammatory reactions[22,23]. Interestingly, bats have evolved immune tolerance not only in response to pathogen challenges, but also to adapt to the cellular and DNA damages induced by the intense metabolism and high body temperatures during flight. Bats have elevated expression of heat shock proteins

(HSPs) to withstand high temperatures and oxidative stress associated with flight. Furthermore, bats have dampened the activity of the pattern recognition receptor STING, lost the PYHIN gene family, and suppressed the NLR-family pyrin domain containing 3 (NLRP3) and the inflammatory cytokine IL-1β, thus curbing the inflammation triggered by flight-induced DNA damage[22]. These traits should also help bats tolerate lethal viral infections, as the damage and death caused by infections are often also closely associated with high fevers and inflammation.

While research on adaptive immunity in bats remains limited, the major histocompatibility complex I (MHC-I) molecule stands out as a subject of interest[24–28]. The MHC-I molecule plays a crucial role in antiviral immunity by presenting antigenic peptides to activate specific CD8$^+$ T cells to clear virus-infected cells, while its downregulation can activate NK cells and trigger inflammatory responses[29–31]. Bats possess distinctive characteristic in their MHC-I molecule, with prevalent 3- or 5-amino acid (3AA or 5AA) insertions at the N-terminal end of the peptide binding groove (PBG), which may generate a unique antiviral immune response.

[1]National Key Laboratory of Veterinary Public Health Security, Key Laboratory of Animal Epidemiology of the Ministry of Agriculture and Rural Affairs, College of Veterinary Medicine, China Agricultural University, Beijing 100193, PR China. [2]Shenzhen Institute of Advanced Technology, Chinese Academy of Sciences, Shenzhen, Guangdong 518055, PR China. [3]Shanghai Zelixir Biotech Company Ltd., Shanghai 200030, PR China. [4]NHC Key Laboratory of Human Disease Comparative Medicine, Beijing Key Laboratory for Animal Models of Emerging and Remerging Infectious Diseases, Institute of Laboratory Animal Science, Chinese Academy of Medical Sciences and Comparative Medicine Center, Peking Union Medical College, Beijing, PR China. [5]The Brain Cognition and Brain Disease Institute, Shenzhen Institutes of Advanced Technology, Chinese Academy of Sciences, Shenzhen, Guangdong, PR China. [6]CAS Key Laboratory of Quantitative Engineering Biology, Shenzhen Institute of Synthetic Biology, Shenzhen Institute of Advanced Technology, Chinese Academy of Sciences, Shenzhen 518055, PR China. ✉e-mail: zhangnianzhi@cau.edu.cn

Peptidomics and crystal structure analyses of Ptal-N*01:01 have shed light on the impact of the 3AA insertion (MDL52-54) on peptide binding[24,25,27]. The 3AA insertion facilitates the formation of a salt-bridge network between the N-terminal end of the PBG of Ptal-N*01:01 and peptides containing aspartic acid as their first residue (P1-D), thereby enhancing their binding. Additionally, Ptal-N*01:01 is capable of binding long peptides of up to 15 amino acids. These studies indicated that the 3AA insertion can augment the binding of peptides to bat MHC-I molecules.

There are still some questions about the insertions in bat MHC-I. First, 5AA insertions are more prevalent in bat MHC-I alleles and their roles remain unknown. Second, the mechanism for how insertion affects bat MHC-I is not fully characterized. Current studies suggested that the effect of the insertion on bat MHC-I depends on the subsequent occurrence of a pair of residues with opposite charges. However, this pair is not present in all bat MHC-I molecules, suggesting that there should be other possibilities. Interestingly, the similar insertions in MHC-I have also been observed in ectothermic animals (bats and marsupials) and reptiles, all which have large fluctuations in body temperature[28,32]. All these insertions, including 5AA insertions, occur within the relatively unstable $3_{10}$ helix of MHC-I, a region prone to conformational changes[33,34]. Based on these observations, it reasonable to speculate that these insertions are associated with the thermal stability of the peptide/MHC-I complex (pMHC-I).

In this study, we uncovered the peptide binding motif (PBM) and structures of the bat MHC-I molecule Mylu-B*67:01 and its mutant Mylu-B*67:01ΔMQQPW, featuring a 5AA insertion (MQQPW52-56). We demonstrated that both 5AA and 3AA insertions can enhance the formation and thermal stability of pMHC-I by stabilizing the $3_{10}$ helix and may help pMHC-I to tolerate the high body temperatures during flight, but the insertion and the inserted bat MHC-I allele need to match each other. Furthermore, AlphaFold2 (AF2) predictions indicated that the effects of different insertions on bat MHC-I were similar to those observed in Mylu-B*67:01 or Ptal-N*01:01. These findings showed that the insertions could improove the thermal stability of bat pMHC-I, and significantly enhance the peptide binding capacity of bat MHC-I, including the quantity, length, and stability of bound peptides. We hypothesized that these inserts serve a dual purposes: first, to help pMHC-I tolerate the high temperatures of bat flight and avoid undue dissociation to cause unwanted immune responses; and second, to improve peptide presentation and clear virus-infected cells more efficiently.

## Results

### The peptide binding motif and crystal structure of Mylu-B*67:01 with MQQPW insertion

The distinguishing feature that sets bat MHC-I molecules apart from those of other mammals lies in the presence of three or five AA insertions within the α1 region of the heavy chain (HC)[26,28]. Among the 131 available bat MHC-I sequences in GenBank, the 5AA insertions were found to be the most prevalent, accounting for 72% (Supplementary Fig. 1a, b). These inserted sequences exhibit notable diversity, with MQQPW, MEGPW, and MEQPW being more commonly observed (Supplementary Fig. 1b). Notably, unlike the 3AA insertions, which are also observed in marsupial and reptile MHC-I molecules, the 5AA insertion appears to be unique to bat MHC-I. In addition, previous studies have shown that the 3AA insertion in the bat Ptal-N*01:01 enhances binding to peptides with aspartic acid as the first residue (P1-D), but requires the participation of aspartic acid at position 59 (D59) and arginine at position 65 (R65) in the formation of a salt bridge network[24,25].

To explore the influence of the 5AA insertion on the structure and function of bat MHC-I, we focused on Mylu-B*67:01. Mylu-B*67:01 contains the most common 5AA insertion sequence, MQQPW, and also exhibits the D61/R67 pairing. The peptide-binding motif (PBM) of Mylu-B*67:01 was determined using our previously established RPLD-MS technique[24]. Initially, Mylu-B*67:01 and bat β2m were co-refolded in vitro with a library of random 9 peptides (R9Ps), followed by purification to obtain the pMylu-B*67:01-R9Ps complexes (Fig. 1a). The

peptides bound within the complexes were eluted with a weak acid, sequenced using the De Novo LS-MS/MS technique, and ultimately, the PBM of Mylu-B*67:01 was visualized using Icelogo[35] (Fig. 1b). The PBM exhibited a clear preference for Mylu-B*67:01 to bind peptides with a proline at the P2 position (P2-P), while hydrophobic amino acids with long side chains, like phenylalanine (F), leucine or isoleucine (denoted as L due to indistinguishability in mass spectra), methionine (M), or valine (V), were predominantly observed at the PΩ position (Fig. 1b). It is noteworthy that D/E occupies a relatively high proportion of the P1 position, although not as pronounced as P2-P. Three peptides matching the PBM were screened and synthesized from the spike protein of COVID-19, and their binding to Mylu-B*67:01 was confirmed through in vitro refolding (Fig. 1c), which validated the accuracy of the obtained PBM. Among these three peptides, P1-FPQSAPHGV exhibited highest refolding efficiency (Fig. 1c) and ultimately successfully formed crystals in complex with Mylu-B*67:01 (pMylu-B*67:01-P1). The crystal structures of the pMylu-B*67:01-P1 complex achieves a resolution of 2.2 Å. Detailed crystal parameters can be found in Table 1.

The crystal structure of pMylu-B*67:01-P1 unveils the intriguing consequences of the MQQPW insertion, forming an additional delicate helix nestled between the $3_{10}$ Helix and α1 helix of HC (Fig. 1d). The inserted MQQPW resides at the junction of the $3_{10}$ Helix and the emerging helix, creating a dainty loop connecting these two helical structures. The MQQPW insertion maintains a considerable distance from the peptide and does not engage in direct interactions (Fig. 1e). The presence of the R67 side chain induces the closure of the A pocket, thereby obstructing the exposure of the P1-F side chain. Notably, despite the close proximity of D61 and R67, no salt bridge was formed between them (Fig. 1e).

The preference for P2-P in the B pocket of Mylu-B*67:01 aligns with observed binding characteristics in other pMHC-I structures (57-61). Residues at positions 71 and 72 emerge as pivotal determinants in shaping the B pocket preference (Fig. 1f). The large aromatic side chain of F/Y72 poses a hindrance to the deep insertion of the P2 residue side chain into the pocket, while I71 establishes a favorable hydrophobic interaction environment. Consequently, P2-P aligns exceedingly well within this B-pocket configuration, emerging as a remarkable anchoring residue. Y7 is highly conserved in the vast majority of MHC-I molecules[36], and the other constituent residues are blocked by F/Y72 from contacting P2 residue, so none of them are determinants of B pocket preference.

The F pocket of Mylu-B*67:01 closely resembles the monkey MHC-I molecule Mamu-A*01:01[37], with a slight difference in a single residue (I/L100). Both molecules show a preference for binding PΩ-L, indicating similar anchoring residue preferences (Fig. 1g). Additionally, the F pocket of Mylu-B*67:01 also demonstrates affinity for other long-side chain hydrophobic residues (F/M/V), similar to the observed binding patterns in pHLA-A24var and pHLA-B*14:02 structures[38,39]. Detailed analysis reveals significant similarity in residues 86, 100, and 121 within the F pocket, along with conserved sites across most MHC-I molecules (89/123/128/129/148/151/152). These sites contribute to shaping the size and phobic landscape of the F pocket, creating an ideal environment for binding PΩ-F/L/M/V.

### The inserted MQQPW enhances peptide binding but is not solely dependent on the assistance of D61/R67

To examine how the insertion influences the peptide presentation by Mylu-B*67:01, a mutant version of Mylu-B*67:01, named Mylu-B*67:01ΔMQQPW, was created by removing the MQQPW sequence. In vitro refolding experiments demonstrated that this mutant could still bind to the COVID-19th P1-P3 peptides, albeit with slightly reduced complex formation compared to the wild-type HC (Supplementary Fig. 2). This difference in complex formation became more pronounced when co-refolding with R9Ps (Fig. 2a). RPLD-MS analysis revealed that the PBM of Mylu-B*67:01ΔMQQPW was similar to that of the wild-type HC (Fig. 2b). However, the preference for P2 and PΩ anchoring residues was significantly diminished, and the difference of P1 residues was relatively obvious (Fig. 2c). Based on previous studies of Ptal-N*01:01, it was hypothesized that the

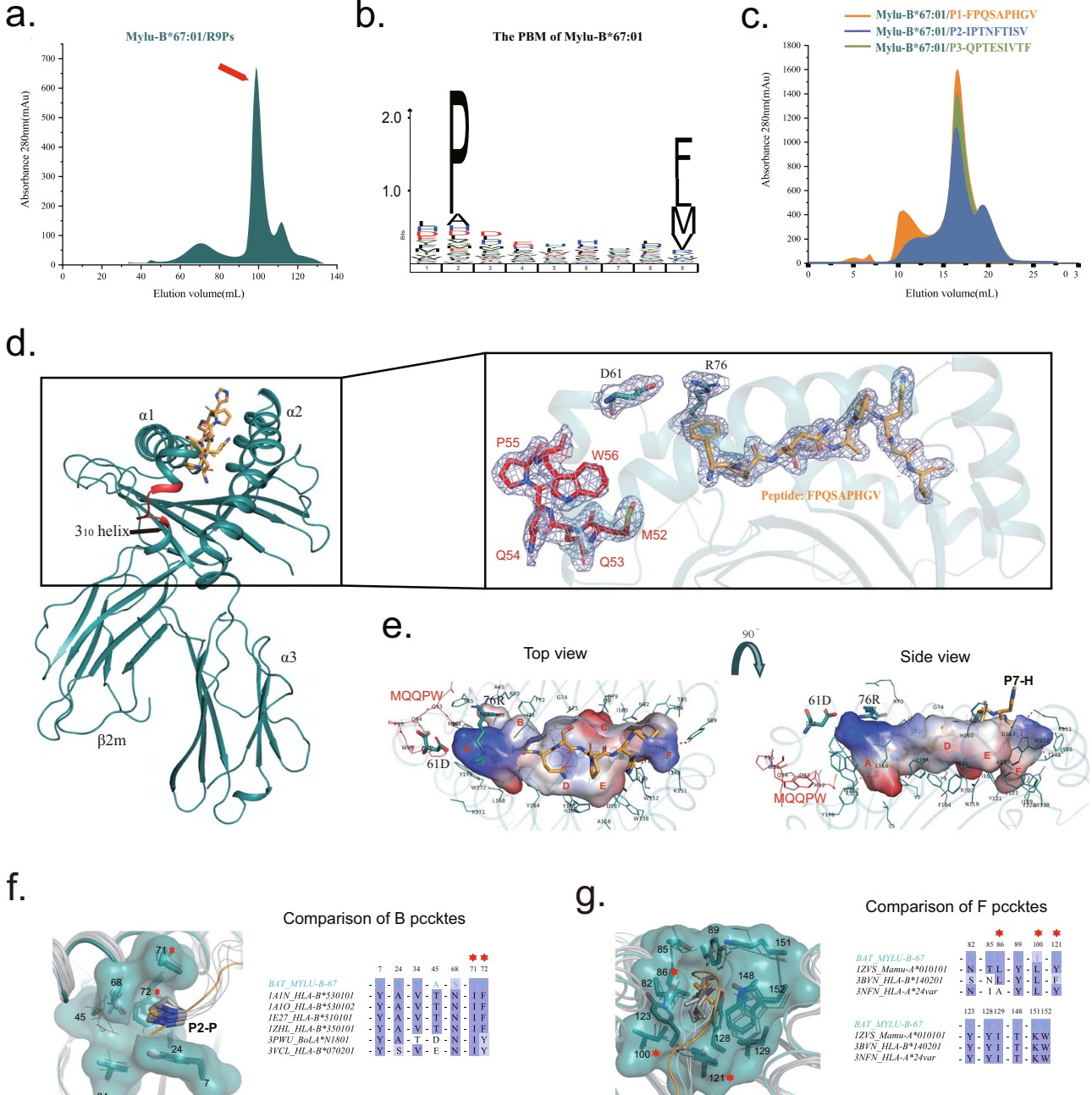

**Fig. 1 | Identification of the PBM of Mylu-B*67:01 and display of the MQQPW insertion pattern and peptide-binding structural basis in the pMylu-B*67:01-P1 structure. a** The in vitro refolding results of Mylu-B*67:01 with the R9Ps are elucidated through gel filtration chromatograms. The chromatographic profile indicates that the peak for the target complex is anticipated to be observed near the 100 ml mark on a Superdex 200 Big Gel Filtration Chromatography column. **b** The PBM of Mylu-B*67:01 determined by RPLD-MS. Sequence logo created using icelogo software shows the AA-weighted probability of binding R9Ps at each position of type Shannon. **c** The in vitro refolding results of Mylu-B*67:01 with three COVID-19 peptides (P1, P2, and P3) as shown in the gel filtration chromatogram. All three peptides were screened from the spike protein based on the PBM of Mylu-B*67:01.The sample destination peak should appear near 16 ml on a superdex 200 small gel filtration chromatography column. **d** The overall structure of pMylu-

B*67:01-P1 and the electron density map highlighting the significant residues and the bound peptide. pMylu-B*67:01-P1 is composed of the alpha chain (HC), β2m and the peptide. The insertion sequence MQQPW is displayed in red, with a black arrow indicating the location of the $3_{10}$ helix. The peptide, inserted MQQPW, D61 and R67 are shown in stick form, and their unbiased 2Fo-Fc electron density maps are also shown. The contour level for the unbiased 2Fo-Fc electron density map is set at 1.0 sigma. **e** Charged surface and residue composition of the PBG of pMylu-B*67:01-P1. Six pockets marked in red with P7-H sticking out of the PBG. D61 and R67 are bolded to indicate their significance. **f** Structure and sequence comparison of the B pocket of Mylu-B*67:01 with other B pockets that prefer to accommodate P2-P, with key P71 and P72 residues marked with red stars. **g** Structure and sequence comparison of the F pocket of Mylu-B*67:01 with other similar F pockets, where key P86, P100 and P121 residues were marked with red stars.

MQQPW insertion might affect the formation of a salt bridge between D59 and R65, consequently impacting the anchoring of P1 residues. To explore this hypothesis, the COVID-19 P1 peptide was mutated at the P1-F position, generating P1-T and P1-E mutants that were more compatible with the P1

residue preference of Mylu-B*67:01ΔMQQPW (Fig. 2d). However, both wild-type and mutant COVID-19 P1 peptides exhibited better binding to Mylu-B*67:01 than Mylu-B*67:01ΔMQQPW (Fig. 2d). The crystal structures of wild-type and mutant P1 peptides in complex with Mylu-

## Table 1 | X-ray diffraction data processing and refinement statistics

| Parameter | MYLU-B-67/P1-F | MYLU-B-67$_{\Delta 5MQQPW}$/P1-F | MYLU-B-67/P1-T | MYLU-B-67$_{\Delta 5MQQPW}$/P1-T | MYLU-B-67/P1-E |
|---|---|---|---|---|---|
| PDB code | 8HSM | 8HSO | 8HSW | 8HT1 | 8HT9 |
| **Data collection** | | | | | |
| Space group | C222$_1$ | C222$_1$ | C222$_1$ | C222$_1$ | P222 |
| Unit cell parameters (Å) | a = 75.392, b = 80.093, c = 144.617 α = 90.00, β = 90.00, γ = 90.00 | a = 76.265, b = 81.832, c = 144.34 α = 90.00, β = 90.00, γ = 90.00 | a = 75.737, b = 80.150, c = 144.243 α = 90.00, β = 90.00, γ = 90.00 | a = 75.744, b = 81.013, c = 143.778 α = 90.00, β = 90.00, γ = 90.00 | a = 64.922, b = 80.429, c = 82.122 α = 90.00, β = 90.00, γ = 90.00 |
| Resolution range (Å) | 50.00-2.18 (2.22-2.18) | 50.00-2.00 (2.03-2.00) | 50.00-2.00 (2.84-2.71) | 50.00-1.98(2.01-1.98) | 50.00-1.98(2.01-1.98) |
| Total reflections | 298,917 | 171,086 | 166,738 | 373,426 | 199264 |
| Unique reflections | 22,370 | 19652 | 21262 | 17684 | 23537 |
| $R_{merge}$ (%)[b] | 10.1 (117) | 19.0 (363.4) | 23.1 (285.4) | 12.2(127) | 12.0(32.5) |
| Avg I/σ (I) | 24.714 (1.875) | 7.071 (0.222) | 6.2(2.625) | 19.636(1.167) | 13.105(7.000) |
| Completeness (%) | 99.90 | 93.5 | 97.4 | 99.4 | 99.7 |
| Redundancy | 12.8 (11.8) | 5.6 (3.6) | 5.8 (4.4) | 12.3 (8.9) | 6.6(6.8) |
| **Refinement** | | | | | |
| Resolution (Å) | 30.00-2.20 | 40.95-2.10 | 30.8-2.03 | 43.89-2.40 | 43.07-2.20 |
| No.reflections | 20091 | 16306 | 18531 | 16764 | 21263 |
| $R_{factor}$ (%)[c] | 22.998 | 22.77 | 23.42 | 20.819 | 21.17 |
| $R_{free}$ (%) | 25.762 | 27.22 | 26.19 | 27.684 | 25.27 |
| **R M S.Deviations** | | | | | |
| Bonds (Å) | 0.010 | 0.007 | 0.007 | 0.008 | 0.010 |
| Angles (°) | 1.633 | 1.572 | 1.527 | 1.565 | 1.631 |
| Average B factor | 37.59 | 43.31 | 49.608 | 48.254 | 35.54 |
| **Ramachandranplot quality** | | | | | |
| Most favored region (%) | 92.13 | 92.02 | 92.11 | 93.62 | 96.33 |
| Allowed region (%) | 7.87 | 7.98 | 7.89 | 6.38 | 3.67 |
| Disallowed region (%) | 0.00 | 0.00 | 0.00 | 0.00 | 0.00 |

[a]Values in parentheses are for highest-resolution shell.

[b]$Rmerge = \Sigma_{hkl}\Sigma_i |I_i(hkl) - \langle I(hkl)\rangle|/\Sigma_{hkl}\Sigma_i I_i(hkl)$, where $I_i(hkl)$ is the observed intensity and $\langle I(hkl)\rangle$ is the average intensity from multiple measurements.

[c]$R = \Sigma_{hkl}||F_{obs}| - k|Fcalc||\Sigma_{hkl}|F_{obs}|$, where $R_{free}$ is calculated for a randomly chosen 5% of reflections and $R_{work}$ is calculated for the remaining 95% of reflections used for structure refinement.

B*67:01 and Mylu-B*67:01ΔMQQPW were solved, except pMylu-B*67:01ΔMQQPW-P1-E which may be attributed to by its low thermal stability (Fig. 2d). The details of these crystal structures were listed in Table 1. The binding patterns of peptides P1-F and P1-T with both Mylu-B*67:01 and the variant Mylu-B*67:01ΔMQQPW remain largely unchanged, with the primary structural variations localized around the region where the MQQPW sequence has been inserted (Fig. 2e). We refined the structures after the deletion of MQQOW and generated unbiased 2Fo-Fc electron density maps to confirm that the structural changes caused by the insertion are credible (Fig. 2f). Clear electron density maps enable us to conduct further comparisons and analyses.

The structures show that Mylu-B*67:01ΔMQQPW has a classical 3$_{10}$ helix conformation, whereas the insertion of MQQPW introduces a new minor helix and slightly delays the formation of the α1 helix after 61D (Fig. 3a). The insertion of MQQPW brings D61 closer to R67 (Fig. 3b), but the salt bridge network is only formed in the presence of P1-E (Fig. 3a). The salt bridge network induced by P1-E makes a significant shift in the minor helix formed by MQQPW (Fig. 3c). The interactions between P1-E and 61D/67 R is similar to that of Ptal-N*01:01 with an inserted MDL (Fig. 3a, d)[24,25]. From the surface pattern, the change in the position of 61D allows it to participate in the formation of the A pocket and to form a salt bridge network with P1-E via 67 R (Fig. 3e). Combined with the PBM and CD results (Fig. 1b, 2d), we believed that 5AA insertion, like 3AA insertion, can also enhance the binding of bat MHC-I molecules to peptides bearing P1 acidic residue with the help of 61D/67 R.

At the same time, some evidence also implied that the impact of MQQPW insertion is not entirely dependent on the salt bridge network formed by 61D/67 R. MQQPW insertion could enhance the binding of peptide P1-F and peptide P1-T to Mylu-B*67:01 (Fig. 2d), but 61D and 67 R do not form salt bridges in their complex structures. (Fig. 3a). The P1-E mutant peptide was less thermally stable with Mylu-B*67:01 compared to the original P1-F peptide (Fig. 2d), suggesting that the acidic residues are not the only preference for the P1 position. This was also supported by the PBM comparison of Mylu-B*67:01 and Mylu-B*67:01ΔMQQPW (Fig. 2c). Additionally, it is worth noting that many bat MHC-I sequences lack the corresponding charged residue pairing after the insertion of 3AA or 5AA (Supplementary Fig. 1c). The pairing of D59 and R65 accounts for 50% of the 3AA insertion sequences. The pairing of this residue combination (D61 and R67) accounts for about 17% of the 5AA insertion sequence, and the pairing of residues with the potential to form a salt bridge accounts for 31% (Supplementary Fig. 1c). Together, these findings indicated that the insertion of 5AA or 3AA in bat MHC-I could also enhance binding to peptides in a manner not involving the 61D/67 R, especially to peptides with non-acidic residues at the P1 position.

### Insertion can enhance the stability of bat pMHC-I by strengthening the 3$_{10}$ helix region

The MQQPW insertion is precisely located within the 3$_{10}$ helix region, which is the most unstable part of the α-helix structure of the MHC-I molecule[33]. The B-factor can be used to characterize the stability of the

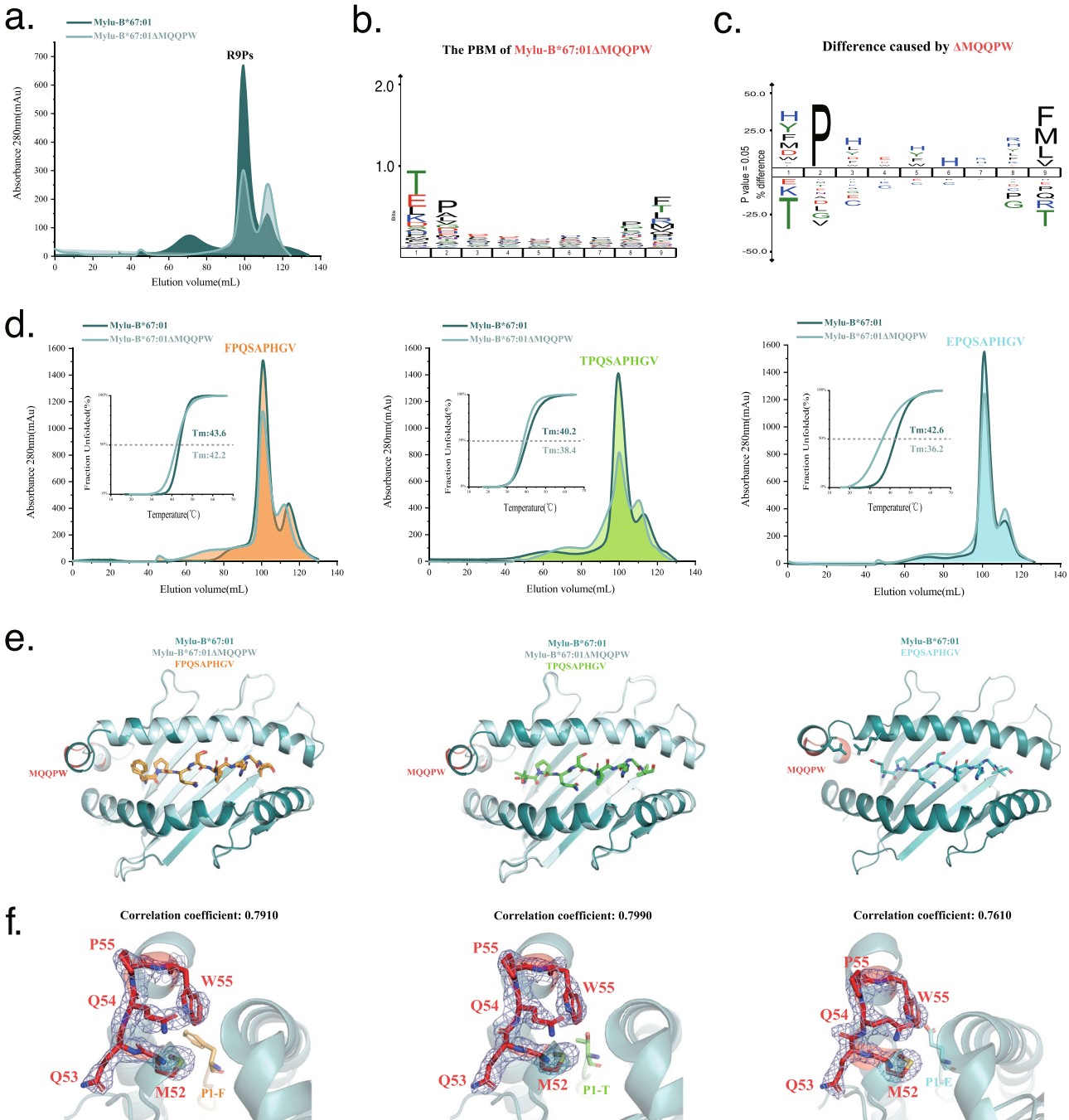

**Fig. 2 | Comparison of Mylu-B*67:01 and Mylu-B*67:01ΔMQQPW peptide binding capacity and crystal structures formed with P1-F/T/E, respectively.** **a** Comparison of the complex formation between Mylu-B*67:01 and Mylu-B*67:01ΔMQQPW with R9Ps through in vitro refolding. **b** The PBM of Mylu-B*67:01ΔMQQPW determined by RPLD-MS. **c** Differential comparison of the PBMs of Mylu-B*67:01 and Mylu-B*67:01ΔMQQPW. **d** Comparison of in vitro refolding products of Mylu-B*67:01 and Mylu-B*67:01ΔMQQPW with P1 peptide and its mutant peptides, respectively. The thermal stabilities of these complexes were also compared by the Tm values determined by CD experiment. **e** Comparison of the solved pMylu-B*67:01 and pMylu-B*67:01ΔMQQPW structures. Peptides P1-F and P1-T bind with Mylu-B*67:01 and Mylu-B*67:01ΔMQQPW in an identical structure. We performed Min-Max normalization of the B factors in the peptide binding regions of Mylu-B*67:01 and Mylu-B*67:01ΔMQQPW to compare the effect of MQQPW insertion on the B factors of the relevant regions (Fig. 4a). In Mylu-B*67:01ΔMQQPW, the 3₁₀ helix region is one of

pattern. The primary structural differences are localized in the region surrounding the insertion of the MQQPW sequence. **f** The unbiased 2Fo-Fc electron density maps of the inserted MQQPW in three solved pMylu-B*67:01 structures (contoured at 1.0 sigma). After the deletion of MQQPW, these three structures were refined using phenix.composite_omit_map, and subsequently, the unbiased 2Fo-Fc electron density maps were displayed using PyMOL to assess the fit of the sticks of MQQPW. The correlation coefficient values between map and model for the inserted MQQPW of the structures are 0.7910, 0.7990, and 0.7610, respectively. The clear matching results substantiate the credibility of the structural changes caused by MQQPW.

the regions with the highest B factors, and in Mylu-B*67:01, the insertion is able to reduce the B factors of neighboring residues. The B-factor putty model can visualize how the B-factors change in the structure, and the comparative analysis revealed that the insertion of MQQPW reduced the

**Fig. 3 | Comparison of the crystal structures of the complexes of wild-type and mutant P1 peptides with Mylu-B\*67:01 and Mylu-B\*67:01ΔMQQPW. The MQQPW insertion was shown in red. a** Hydrogen bond analysis plot of the key amino acids D61 and R67 (positions in the structure of pMylu-B\*67:01-P1-F as an example) with the P1 residue of the binding peptide in different bat MHC-I structures. The insertions are highlighted in red. D61, R67 and the P1 residue of the peptide are shown in stick form. In the structure of pMylu-B\*67:01-P1-E, D61 and R67 form a salt-bridge network with residue E in the first position of peptide, consistent with the salt-bridge network formed at the corresponding position in pPtal-N\*01:01. **b** Merged comparison of the structures of pMylu-B\*67:01-P1-F/T and pMylu-B\*67:01ΔMQQPW-P1-F/T. In the structures that exhibit the MQQPW insertion, D61 is closer to the 67 R and P1 residues than the corresponding D57 in the structures without the inserted sequence. **c** Merged comparison of pMylu-B\*67:01-P1-F/T/E. As indicated by the red arrow, the presence of a salt-bridge network in the structure of pMylu-B\*67:01-P1-E results in a large shift in the small helix following the inserted MQQPW. **d** The binding mode of P1-E to Mylu-B\*67:01 resembles that of P1-D to Ptal-N\*01:01. **e** The relative position between D61 and R67 is changed by the insertion of MQQPW, allowing them to form a salt bridge network with P1-E.

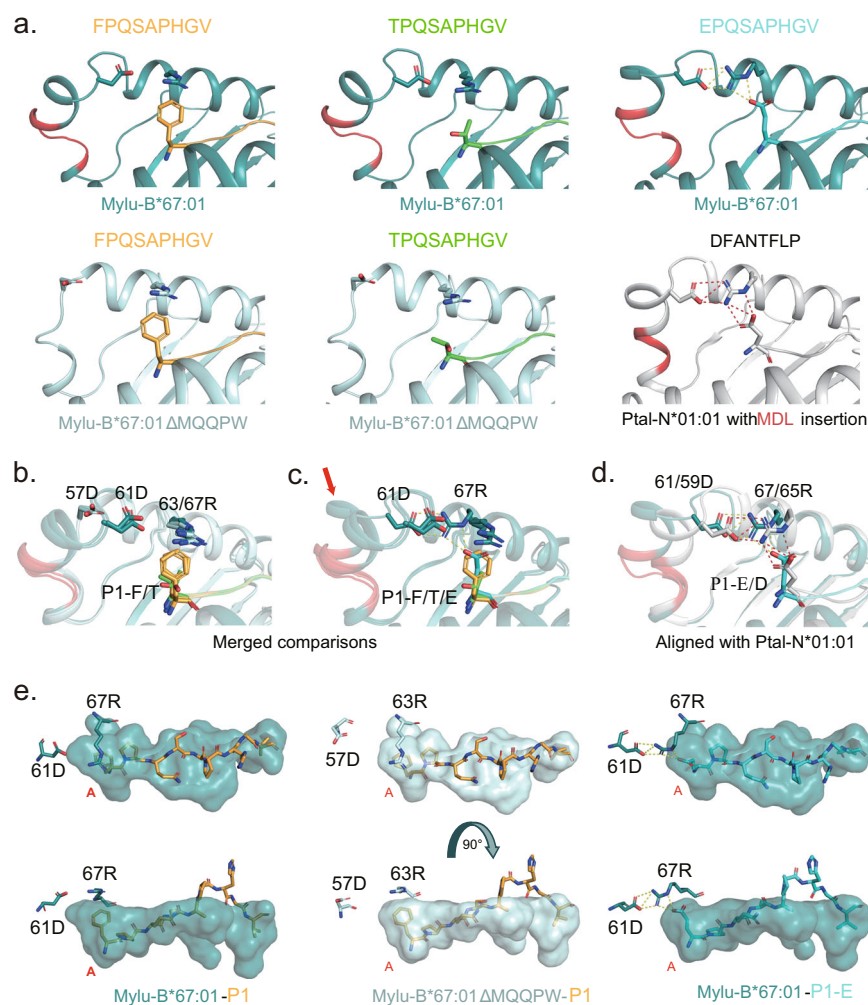

B-factor of the 3₁₀ helix region, indicating an improvement in the structural stability of this region (Fig. 4b). This was further supported by the analysis of hydrogen bonding in this region (Fig. 4c). In Mylu-B\*67:01, there are a total of 12 intra-chain hydrogen bonds in this region to maintain the structure, whereas in Mylu-B\*67:01ΔMQQPW, there are only four hydrogen bonds. The conformation of W52 changes significantly during peptide exchange[34], suggesting that improving the stability of W52 contributes to the stabilization of the complex. Within the helix formed by the MQQPW insertion, the inserted W58 forms a hydrogen bond with W52 on the 3₁₀ helix, which helps to strengthen the link between the two smaller helices.

Subsequently, we examined whether the 3AA insertion had a similar effect on Ptal-N\*01:01. Their B factors in the peptide binding regions were compared in (Fig. 4d, e). The analysis revealed that the MDL insertion, similar to MQQPW, reduce the B factor of the 3₁₀ helix region, but through a different mechanism. The insertion of MDL extends the 3₁₀ helix and forms 6 hydrogen bonds to stabilize the structure, which is four more than in the ΔMDL mutant. (Fig. 4f). The inserted L55 forms a hydrogen bond with W52 on the 3₁₀ helix. So we believed that both 3AA and 5AA insertions can enhance the stability of bat pMHC-I by strengthening the 3₁₀ helix region, independent of D61/R67.

**Insertion may help bat pMHC-I to tolerate high body temperature during flight and improve the peptide binding ability, but requires a match to the inserted allele**

Both 3AA and 5AA insertions have been shown to enhance the thermal stability of bat pMHC-I with specific peptides[25] (Fig. 2d). However, relying solely on the thermal stability measurements of a few pMHC-I individuals

may not adequately reflect the overall impact of the insertion. To address this, we assessed the thermal stabilities of mixed pMylu-B\*67:01-R9Ps and pMylu-B\*67:01ΔMQQPW-R9Ps. The circular dichroism (CD) analysis revealed that the insertion of MQQPW increased the Tm value of the mixed pMHC-Is by approximately 6.5 °C (Fig. 5a), which closely matches the magnitude of temperature increase observed during bat flight. We further examined the changes in these mixed bat pMHC-Is after incubation at the body temperature experienced by bats during flight (41 °C) for 30 min and 60 min (Fig. 5b). As expected, 41 °C exceeded the Tm value of the mixed bat pMHC-Is, resulting in the dissociation of more than half of the pMHC-Is, consistent with the CD results. However, at both time points, a higher proportion of pMylu-B\*67:01-R9Ps remained compared to pMylu-B\*67:01ΔMQQPW-R9Ps, indicating that the insertion of MQQPW indeed improves the tolerance of bat pMHC-Is to high body temperature. Moreover, the thermal stability of MHC-I molecules plays a crucial role in peptide presentation[40,41]. The insertion of MQQPW leads to structural solidification, thereby enhancing the peptide-binding capacity of bat MHC-I. In vitro experiments demonstrated that Mylu-B\*67:01 exhibited a significantly higher binding ability for both short and long peptides compared to Mylu-B\*67:01ΔMQQPW (Fig. 5c).

The insertion of MDL also enhanced the thermal stability of pPtal-N\*01:01-R9Ps and improved its tolerance to 41 °C, although to a lesser extent compared to MQQPW (Fig. 5d, e). The MDL insertion also significantly increased the number and length of peptides that Ptal-N\*01:01 could bind (Fig. 5f).

Previous studies have indicated that the insertion of MDL does not enhance the peptide-binding capacity of HLA-A\*02:01[25]. Additionally,

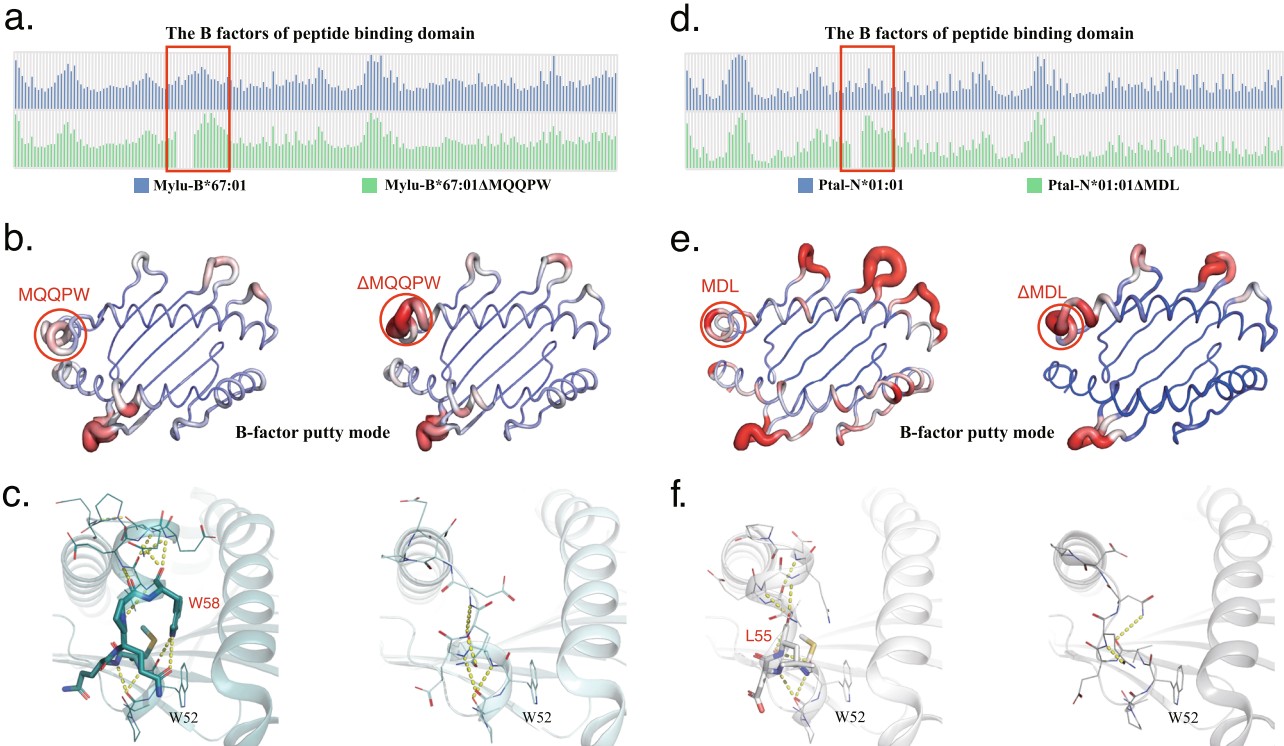

**Fig. 4 | The MQQPW/MDL insertion Enhances Stability of the 3₁₀ Helix Region.** **a** The results of Min-Max normalization of the B factors in the peptide binding regions (P1-P180) of pMylu-B*67:01 and pMylu-B*67:01ΔMQQPW, the area pMylu-B*67:01 is shown as a blue bar graph and pMylu-B*67:01ΔMQQPW is shown as a green bar graph. The height of the bar graph represents the magnitude of the B factor; the larger the B factor, the greater the diffusion of the electron density map in the corresponding region and the more unstable the conformation. **b** The B-factor putty model of pMylu-B*67:01 and pMylu-B*67:01ΔMQQPW, with the insertion sequence MQQPW and its associated regions indicated by red circles. This region of pMylu-B*67:01 has a relatively low B-factor value compared to pMylu-B*67:01ΔMQQPW. In the B-factor putty model, the lines' color and thickness represent the magnitude of the B factor. The transition from blue to red represents the gradual increase of the B factor, and the thicker the line represents the larger the B

factor. **c** Hydrogen bond analysis of the region around 310 helix in the structures of pMylu-B*67:01 and pMylu-B*67:01ΔMQQPW. The inserted MQQPW introduces more hydrogen bonds in this region. **d** The results of Min-Max normalization of the B factors in the peptide binding regions of pPtal-N*01:01 and pPtal-N*01:01ΔMDL, the area marked by the red box is the insertion sequence MDL and its associated region. pPtal-N*01:01 is shown as a blue bar graph and pPtal-N*01:01ΔMDL is shown as a green bar graph. **e** The B-factor putty model of pPtal-N*01:01 and pPtal-N*01:01ΔMDL, with the insertion sequence MDL and its associated regions indicated by red circles. This region of pPtal-N*01:01 has a relatively low B-factor value compared to pPtal-N*01:01ΔMDL. **f** Hydrogen bond analysis of the region around 310 helix in the structures of pPtal-N*01:01 and pPtal-N*01:01ΔMDL. The inserted MDL introduces more hydrogen bonds in this region.

insertions in bat MHC-I alleles exhibit different sequences and lengths (Supplementary Fig. 4a, b), suggesting that the effects of insertions may be specific to certain species or alleles. To investigate this hypothesis, we selected the bat MHC-I allele Mylu-B*57:01, which does not possess any amino acid insertion. We introduced MDL and MQQPW insertions into Mylu-B*57:01 and compared their impact on peptide binding (Fig. 5g). The results demonstrated that only MDL was able to enhance the binding of Mylu-B*57:01 to random nonapeptides and slightly improve the thermal stability of pMHC-I. In contrast, MQQPW did not exhibit any enhancing effect and significantly reduced the thermal stability of pMHC-I (Fig. 5h).

These findings indicated that both the 3AA and 5AA insertions may enhance the ability of bat pMHC-I to tolerate high body temperature during flight and improve the peptide binding, but the insertion and the inserted bat MHC-I allele need to match each other.

**Analysis of the impact of diverse insertions on the structure of bat MHC-I with AlphaFold2**

Investigating the effect of diverse insertions observed in numerous bat MHC-I alleles by crystallographic methods, as done for Mylu-B*67:01 and Ptal-N*01:01, is an impractical approach. The emergence of AlphaFold2 (AF2) has significantly enhanced the accuracy of structure prediction[42], offering a potential solution to solve this problem. Prior to AF2, predictive methods like Swiss Model could predict the structure of MHC-I, but their accuracies was limited by sequence similarity between the target sequence

and template, particularly for MHC-I molecules with substantial sequence variations across species, such as Mylu-B*67:01 (Supplementary Fig. 3). Moreover, previous tools struggled to accurately predict the complex structure of pMHC-I, particularly the peptide-MHC-I binding. DeepMind, utilizing AF2, has developed a novel tool called AF-Multimer[43] to predict complex structures. The main question to address is whether AF2 and AF-Multimer can effectively predict the structure of pMHC-I, particularly in unknown animal pMHC-I. We utilized AF2 to predict the structures of Mylu-B*67:01 HC and Mylu-B*67:01ΔMQQPW HC, as well as the structures of pMylu-B*67:01 and pMylu-B*67:01ΔMQQPW using AF-Multimer (Fig. 6a). AF2 accurately predicted the formation of the additional small helix by MQQPW insertion with a high confidence level (pLDDT > 90) comparable to that of the rest of the structure. The predicted structures closely resemble the actual structures (Fig. 6b), exhibiting RMSD values of 0.723 Å for pMylu-B*67:01 and 0.707 Å for pMylu-B*67:01ΔMQQPW. Remarkably, AF-Multimer's prediction of the peptide binding conformation closely matched the true conformation. The above results indicated that AF2 can be used to predict the structure of bat MHC-I. Consequently, we employed AF2 to investigate the various AA insertions in bat MHC-I alleles (Fig. 6c). The predicted structures of diversve 5AA insertions and 3AA insertions are similar to MDL and MQQPW respectively.

Sequence alignment revealed that among the 5AA insertions, W58 was highly conserved, except in four sequences (XP_0244236.1/ XP_036985990.1/XP_037023937.1/XP_0194562) (Supplementary Fig. 4).

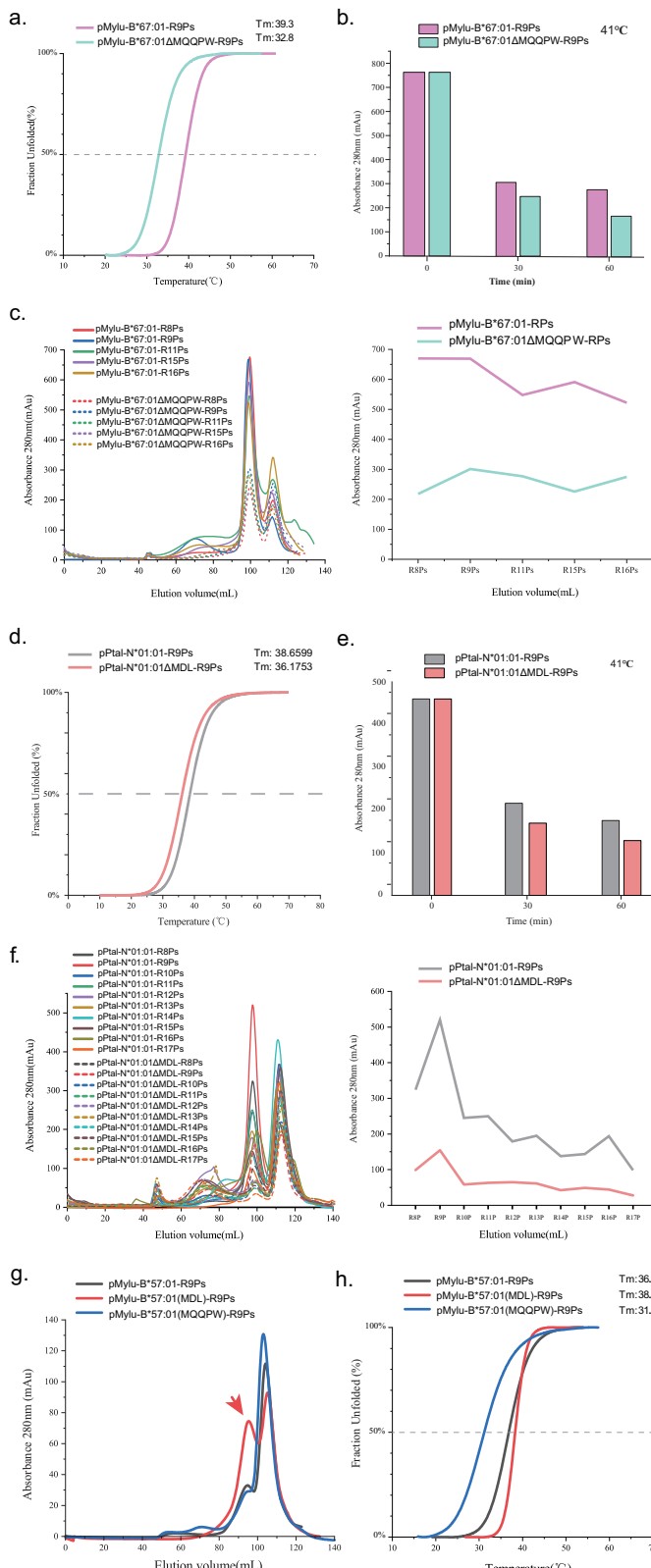

**Fig. 5 | In suitable alleles, insertion sequences can enhance their peptide binding capacity and promote tolerance to high temperatures. a** Thermal stabilities of pMylu-B*67:01-R9Ps and pMylu-B*67:01ΔMQQPW-R9Ps as shown by CD results. **b** The MQQPW insertion enhanced the tolerance of pMylu-B*67:01-R9Ps to 41°C. The vertical coordinate is the UV absorption value of the complex peak in the gel filtration chromatogram. **c** MQQPW insertion significantly increased the number of peptides with different lengths bound by Mylu-B*67:01. The line graph on the right represented the UV absorption value of the complex peak in the gel filtration chromatogram. **d** Thermal stability of pPtal-N*01:01-R9Ps and pPtal-N*01:01ΔMDL-R9Ps as shown by CD results. **e** The MDL insertion enhanced the tolerance of pPtal-N*01:01-R9Ps to 41°C. **f** MDL insertion significantly increased the number of peptides with different lengths bound by Ptal-N*01:01. The line graph on the right represented the UV absorption value of the complex peak in the gel filtration chromatogram. **g** Differential effects of MDL and MQQPW on the binding capacity of the insertion-free bat MHC-I molecule Mylu-B*57:01 to R9Ps. **h** Differential effects of MDL and MQQPW on the thermal stabilities of the insertion-free pMylu-B*57:01-R9Ps.

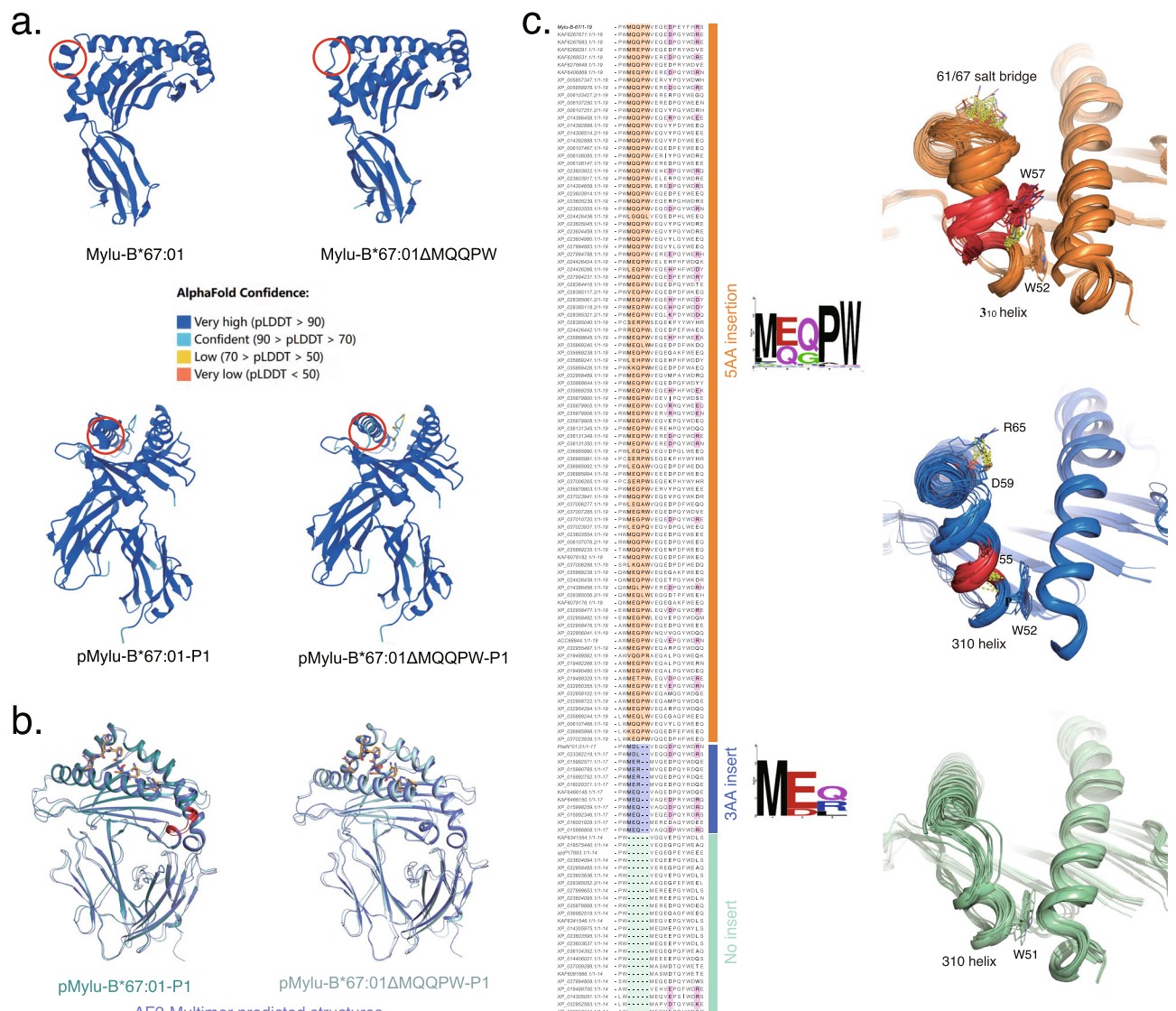

**Fig. 6 | Analysis of polymorphic insertions in bat MHC-I based on AF2 structure prediction. a** Structural prediction of Mylu-B*67:01, Mylu-B*67:01ΔMQQPW, and their complexes with P1 peptide by AF2 and AF2-Multimer. **b** Precise prediction of the structures of pMylu-B*67:01-P1 and pMylu-B*67:01ΔMQQPW-P1 by AF2-Multimer. The predicted structures showed good consistency with the resolved crystal structures, especially regarding the conformation of both the MQQPW insertion and the P1 peptide. **c** Comparison of amino acid sequences near insertion and AF2 prediction of existing bat MHC-I structures. Sequences with five, three, and zero amino acid insertions were represented by orange, blue, and cyan parts, respectively. The pairs of positively and negatively charged residues that appear after insertion were marked in pink. The sequence logos showed the polymorphism of the insertion sequences.

However, predictions indicated that the α2 helix of these four bat MHC-Is varied significantly from the classical MHC-I structure, making the accuracy of their sequences uncertain (Supplementary Fig. 4). The remaining 5AA insertions could form a secondary helix and stabilize the $3_{10}$ helix through a hydrogen bond between W58 and W52, similar to Mylu-B*67:01 (Fig. 6c). Furthermore, all 3AA insertions could extend the $3_{10}$ helix, and the 55th residue could maintain the helix by forming a hydrogen bond with W52, as observed in the case of Ptal-N*01:01 (Fig. 6c). The predictions also indicated that salt bridges could form when residues with opposite charges were paired at positions 61/67 (5AA insertion number) or 59/65 (3AA insertion number). Therefore, we suggested that despite sequence differences, insertions of the same length have similar effects on bat MHC-I and that all insertions should have the ability to enhance $3_{10}$ helix stability.

## Discussion

The MHC-I molecule plays a key role in antiviral immunity, and the presence of prevalent and polymorphic 5AA or 3AA insertions in bat MHC-I

molecules has attracted particular attention[32]. However, existing studies have examined 3AA insertions with Ptal-N*01:01 and have not addressed 5AA insertions, which are unique and more prevalent in bat MHC-I molecules. Here, we investigated the 5AA insertion with Mylu-B*67:01. The MQQPW insertion introduces a new small helix and pushes D61 closer to R67, allowing them to form a salt bridge network with the P1 acidic residue of the bound peptide (Fig. 3). This is similar to the effect of MDL insertion on Ptal-N*01:01 (Fig. 3d), suggesting that with the help of a specific pair of oppositely charged residues after insertion, both 5AA and 3AA insertions enhance the binding of bat MHC-I to peptides bearing P1 acidic residues[24,25].

Nevertheless, the above findings do not fully explain the effects of insertion on bat MHC-I. First, the MQQPW insertion also enhances the binding of Mylu-B*67:01 to the peptides with P1 non-acidic residues, but no salt bridge is formed between 61D and 67 R (Figs. 2, 3 and Supplementary Fig. 2). Second, the oppositely charged residue pairing is not present in all bat MHC-I sequences with the insertion (Supplementary Fig. 1c). By comparing the insertion-induced structural changes of bat pMHC-I with

the B factor putty mode, a presentation that indicates structural stability, we found a possible new explanation. Both 5AA and 3AA insertions can reduce the B factors of the $3_{10}$ helix region (Fig. 4), which is an unstable region of the antigen-binding domain that needs to flip its conformation during peptide assembly[33,34]. The 3AA insertion extends the $3_{10}$ helix and increases its stability, while the 5AA insertion forms an additional small helix that links to the $3_{10}$ helix. Both insertions bring more intra-chain hydrogen bonds and interact with W52 on the $3_{10}$ helix, which undergoes a significant deflection during peptide assembly[34]. We believed that these interactions reduce the conformational flexibility of the $3_{10}$ helix and increase the difficulty of peptide dissociation, thus enhancing the thermal stability of bat pMHC-I. The new interpretation we proposed here is more general, as it not only involves the insertion itself and applies to all binding peptides.

The structural stability of bat pMHC-I is strengthened by the insertion, which should be reflected in its increased thermal stability. This can be determined by in vitro thermal incubation experiments, and the determination of mixed complexes formed by MHC-I with multiple peptides can avoid bias due to the use of a few specific peptides and better reflect the overall differences. Jappe, E. C. et al. previously used mixed pMHC-Is captured from cells[40], whereas we used mixed complexes of MHC-I with random peptides formed by in vitro refolding. The results showed that either 3AA or 5AA insertion can improve the thermal stability of bat pMHC-I (Fig. 5a, d) and reduce degradation upon incubation at 41 °C (Fig. 5b, e), which is the body temperature that bats can reach in flight We also confirmed that the mixed pMHC-Is can be used to obtain definite Tm values by CD assay. This will help to assess the overall peptide binding capacity of MHC-I, especially when studying MHC-I molecules in animals with limited conditions.

In addition to enhancing the binding of peptides with P1 acidic residues, the insertions also enhance the ability of bat MHC-I molecules to present diverse peptides. Deleting the insertion reduces the number of random peptides that can bind to Mylu-B*67:01 or Ptal-N*01:01 (Fig. 5c, f). This insertion does not affect pockets of anchoring residues, such as the B and F pockets, but can prevent peptide dissociation by strengthening the $3_{10}$ helix region, thereby increasing the overall number of bound peptides. More eluted peptides helps the identification of the PBM of Mylu-B*67:01 by RPLD-MS (Fig. 1a). The peptidome eluted from the cells showed that Ptal-N*01:01 can present peptides up to 15 AA[24]. Our in vitro results showed that Ptal-N*01:01 could bind more 12-16AA peptides than the deletion mutant (Fig. 5f). Interestingly, Mylu-B*67:01 might have a strong binding capacity for long peptides, as the number of bound long peptides did not drop significantly compared to 9AA peptides (Fig. 5c). We speculated that the insertion improves the peptide binding ability of bat MHC-I molecules and helps bats produce effective antiviral T-cell responses to clear viral infections.

The wide diversity of insertions in bat MHC-I molecules (Supplementary Fig. 1) and their different effects on the same bat MHC-I molecule (Fig. 5g, h) suggest that the insertions and the inserted bat MHC-I have undergone selection during evolution. Recent advances in protein structure prediction allowed us to study diverse insertions on a large scale. Using AF2 and AF2-Multimer, which can accurately predict the structure of pMylu-B*67:01, we predicted the structure of all available bat MHC-I molecules. We did not used the structures of Ptal-N*01:01 and Mylu-B*67:01 to train AF2, but the predicted structures show that all diverse insertions can stabilize the $3_{10}$ helix, similar to Ptal-N*01:01 and Mylu-B*67:01. AF2 and AF2-Mulimer have shown unique advantages in the study of HLA[44,45], and our study suggested that they can also be applied in the study of MHC-I structure in unknown animals to help identify epitopes and develop vaccines.

The reason why bats carry multiple viruses (e.g., coronaviruses) but do not develop the disease may be due to the ability of their immune system to enhance the host defense response while suppressing excessive inflammatory responses caused by the intense physiological metabolism and high temperatures during flight. Some immune molecules associated with inflammation, such as STING, NLRP3, and IL-1β, are dampened or suppressed so as not to react to the damage caused flight. On the other hand, high fever and a strong inflammatory response are important factors in the pathological damage caused by viral infections, so bats are also very resistant to viral infections. It is an interesting question how the bat immune system adapts to flight and affects the antiviral response. Our study suggested that the insertions of MHC-I maybe related to this issue. We proposed the following hypothesis (Fig. 7): To adapt to the extreme temperature fluctuations experienced during flight, bat MHC-I molecules have developed additional insertions at the $3_{10}$ helix. These insertions serve to enhance the stability of this region, which is prone to conformational changes. By preserving the stability of pMHC-I, these insertions may help prevent unnecessary inflammation, such as the activation of NK cells triggered by pMHC-I degradation[29-31]. These insertions not only contribute to the stability of the complex during flight but also enhance the antigen presentation ability of bat MHC-I molecules, which may result in a stronger T-cell immune response and improved clearance of viral infections. The dual role of the insertion in bat MHC-I appears to be in line with the features of the bat immune system and favors bats carrying viruses without getting sick. However, further research is needed to fully understand the implications of these insertions and their role in bat immune responses.

In fact, the 3AA insertion is also present in MHC-I molecules of lower animals such as reptiles and marsupials, and the effect on peptide binding is similar to the case of Ptal-N*01:01[25]. Bats and marsupials are heterotherms, and reptiles are ectotherm, which is less able to regulate body temperature than birds and higher mammals, and therefore have a great variation in body temperature. The presence of insertions in their MHC-I enhances the thermal stability of pMHC-Is, probably to adapt to the large changes in body temperature and to maintain the stability of MHC-I molecular function. To date, the 5AA insertions have been only found in bat MHC-Is, which are more prevalent and highly diverse compared to the 3AA insertions (Supplementary Fig. 1). In suitable inserted bat MHC-I, 5AA insertion may have a stronger enhancement on the thermal stability of pMHC-I (Fig. 5a) compared to insertion of 3AA (Fig. 5d). Bats, due to the intense temperature rise during flight, may had experienced stronger selection pressure than other lower animals, leading to 5AA insertions and higher diversity. These implied that the insertions in the MHC-Is of lower vertebrates might have evolved in response to their drastic changes in body temperature.

## Methods
### Sequence retrieval analysis and synthesis
The sequences of 131 MHC-I (including predicted genes) of bats were retrieved from the NCBI database (https://www.ncbi.nlm.nih.gov/) (Supplementary Table 1), and the sequences of human MHC-I were retrieved from the Immunopolymorphism Database (IPD) (www.ebi.ac.uk/ipd/mhc), and sequence comparisons were performed using MEGA-X[46] and Jalview[47].

Among the MHC I sequences used in this paper, the extracellular region expression plasmids of Ptal-N*01:01 (GenBank No. KT987929), Ptal-N*01:01ΔMDL, and bat β2m (GenBank No. XP_006920478.1)[24] was previously constructed in our laboratory. The amplification product of the extracellular structural domain of Mylu-B*67:01 (GenBank number: XP_006107180.2), Mylu-B*57:01(GenBank number: XP_023604095.1) was cloned into the pET-21a vector (Novagen). To investigate the functions of Met52, Gln53, Gln54, Pro55, and Trp56, overlapping PCR was used (using the following primers:5'-TGCTCCTTGGGGTTGAACAGGAAGATCCG GAATA-3',5'-TGTTCAACCCAAGGAGCACGAGGTTCGGCTTTAG G-3'.), the five amino acid deletion mutant sequence Mylu-B*67:01ΔMQQPW was constructed and cloned into the pET-21a vector (Novagen).To investigate the effect of three and five amino acid insertions on bat MHC I molecules, we constructed two mutant sequences: MYLU-B-57-3 and MYLU-B-57-5 using overlapping PCR(using the following primers: MYLU-B-57-3:5'-GGATGGATTTAATG GAAAGAGAGGAACCGGGTTA-3',5'-TTCCATTAAATCCATC-CAAGGGGCACGTGGCT-3',MYLU-B-57-5:5'-ATGCAGCAGCCGTG-GATGGAAAGAGAGGAACCGGGTTA-3',5'- ATCCACGGCTGCTGC

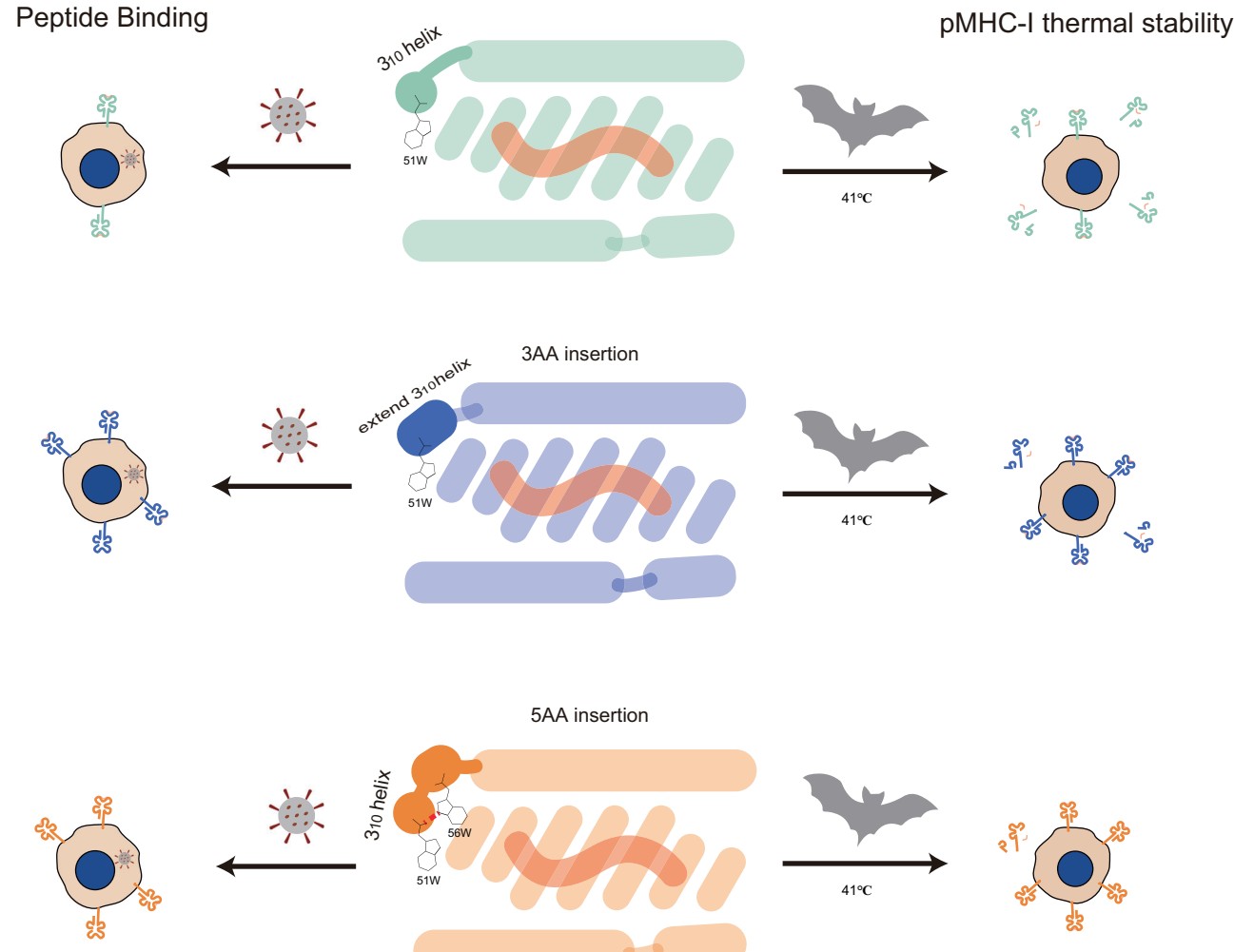

**Fig. 7 | The hypothesis of a dual effect of insertion in MHC-I on bat immunity.** Schematic diagrams showing the antigen peptide presentation ability of bat MHC-I molecules with 0AA insertion, 3AA insertion, and 5AA insertion, as well as the thermal stability of antigen peptide binding to MHC-I molecules at a bat body temperature of 41 °C. In the schematic diagram of 5AA insertion, more hydrogen bonds can be seen.

ATCCAAGGGGCACGTGGC −3'.) and cloned them into the pET-21a vector (Novagen).

## Peptide synthesis

The peptides with known sequences used in this experiment are listed in Supplementary Table 2. Eleven synthetic random peptide libraries, Ran_7Xsplitted -Ran_17Xsplitted (where X is a random amino acid other than cysteine). Synthesize as before to ensure its randomness[48]. All peptides were synthesized and purified to 99% by reverse-phase high-performance liquid chromatography (HPLC) and mass spectrometry (SciLight Biotechnology). All peptides are stored as a lyophilized powder in a −80 °C refrigerator and dissolved in dimethyl sulfoxide (DMSO) before use.

The constructed plasmids were transformed into E. coli strain BL21 (DE3). After prokaryotic expression of the plasmid and purification to inclusion bodies, the purified protein was dissolved in 6 M guanidine hydrochloride at a final concentration of 30 mg/ml and stored at −20 °C as described previously[49,50]. MHC heavy chain, β2m, and peptide were added slowly dropwise to the refolding buffer (100 mM Tris-HCl (pH 8.0), 2 mM EDTA, 400 mM L-Arg-HCl (JS0276, HongKong JiSiEnBei International Trade Co., Limited), 0.5 mM oxidized glutathione and 5 mM reduced glutathione) in a molar ratio of 1:1:3. After proteins were magnetically stirred in a revival buffer system for 24 h at 4 °C, the complexes were concentrated and exchanged into a buffer of 20 mM Tris-HCl (pH 8.0) and 50 mM NaCl, and the complexes were further purified by gel filtration chromatography and anion-exchange chromatography (Superdex 200 16/60 column and Resource Q anion exchange column, GE Healthcare)[51].

## Isolation and Identification of High-Affinity Peptides

The peptide-containing fraction of the complex was acidified with 0.2 N acetic acid at 65 °C for 30 min as described previously[24,51]. Then, the peptide-containing fraction was concentrated through a 3 kDa filter and desalted using a desalting column (WAT094225, waters). The desalting procedure is shown below: 1. The column was activated twice using 200 μl of 0.5% TFA, and 80% ACN 2. Equilibrate the column twice with 200 μl of 0.5% TFA. 3. Replace the collection tube, load the sample onto the column, and repeat. 4. Wash the sample with 200 μl of 0.5% TFA. 5. Replace the collection tube and elute the sample from the column with 200 μl of 0.5% TFA, 80% ACN. 6. Spin dry and store for use.

The EasyNano LC 1000 system (Thermo Fisher Scientific, San Jose, CA) was used to fractionate the peptides. Firstly, the desalted peptides were loaded onto an in-house constructed trap column (5-μm pore size, 150-μm i.d. × 3-cm length, 120 Å) before being separated on a custom-made C18 column (3-μm pore size, 75-μm i.d. × 15-cm length, 100 Å) with a flow rate of 450 μl/min. A linear gradient spanning 60 min was performed, starting with 3% B (0.1% formic acid in acetonitrile [v/v]/97% A (0.1% formic acid in H2O [v/v]) and increasing to 6% B in 8 min, 22% B in 37 min, 35% B in 8 min, and 100% B in 2 min before holding at 100% B for 5 min. The Q Exactive HF (Thermo Fisher Scientific, Bremen) acquired the MS data in

data-dependent acquisition mode, with the top 20 precursors by intensity from mass range m/z 300 to 1800 being sequentially fragmented using higher-energy collisional dissociation at a normalized collision energy of 27. A dynamic exclusion time of 20 s was used, with automatic gain control set to 3e6 and 1e for MS1 and MS2, respectively, and a resolution of 120 and 30 K for MS1 and MS2[24,51].

Peptides were isolated from each spectrum (FDR = 1%) via de novo sequencing in Peaks Studio software based on the spectrum information. The following parameters were set: enzyme was set to nonspecific, variable modifications of oxidation (M)/deamidation (N, Q) were adjusted, peptide mass tolerance was approximately ±10 ppm, and fragment mass tolerance was set to 0.02 Da. After being adjusted by the detection threshold (score ≥ 50). All the data were deposited and conformed to community standards[52]. All nine-amino acid peptides were selected, and their binding motifs were mapped using Icelogo software. The appropriate antigenic peptide in the spike protein of COVID-19 was selected based on the obtained peptide binding motifs.

### Determination of protein thermostability using CD spectroscopy
The sample concentration of all complexes to be measured was adjusted to 0.2 mg/ml.CD spectra at 218 nm were measured on a Chirascan spectrometer (Applied Photophysics) using a thermostatically controlled cuvette at temperature intervals of 0.2 °C at an ascending rate of 20 to 80 °C. The spread fraction (%) is expressed as $(\theta - \theta a)/(\theta a - \theta b)$, where $\theta a$ and $\theta b$ are the average residual ellipticity values in the fully folded and fully expanded states, respectively. The degeneracy curves were generated by nonlinear fitting with OriginPro 8.0[53] (OriginLab). Tm was calculated by fitting the data to the degeneracy curves and using the deformation-determining derivatives.

### Determination of protein thermostability using an in vitro heating method
At the end of the refolding of the p/MHC I molecules, the complexes were purified in the first step using a gel filtration chromatography column (GE), the target complex molecules were collected and the samples were concentrated to 2 ml using a 3 kDa ultrafiltration concentration tube. Each sample was divided into equal triplicates and incubated at 41 °C for 0 min, 30 min, and 60 min, respectively, and then subjected to gel filtration chromatography, and the harvested target complex molecules were quantified.

### Crystallization, data collection, and processing
Three peptides, FPQSAPHGV/TPQSAPHGV/EPQSAPHGV, were utilized to form crystals with Mylu-B*67:01/Mylu-B*67:01ΔMQQPW and β2m, respectively. The crystals were initially screened using the sitting-drop method and the Crystallization Screening Kit (Hampton Research). Purified protein complexes were diluted to 2 and 4 mg/ml with molecular sieve buffer and then mixed with the crystallization screening kit buffer in a 1:1 volume ratio. pMylu-B*67:01/P9-FPQ, pMylu-B*67:01/P9-EPQ, and pMylu-B*67:01/P9-TPQ were grown as relatively thin slices in 0.2 M ammonium acetate, 0.1 M BIS-TRIS pH 7.1, and 20% w/v polyethylene glycol 3350. pPtal-N*01:01ΔMDL/HEV-1 were grown as relatively thick slices in 0.2 M sodium citrate tribasic dihydrate, 0.1 M BIS-TRIS pH 8.3, and 20% w/v polyethylene glycol 3350. Diffraction data were collected at Beamline BL18U1/BL19U of the Shanghai Synchrotron Radiation Facility using an R-AXIS IV + + imaging plate detector. The collected diffraction data were indexed and integrated using iMosflm[54], and scaling and merging were performed using the CCP4i suite[55].

The collected data were processed using CCP4i to determine the structure and calculate the phase via MOLREP[56] and phasers[57] (employing the molecular substitution method[58] and using Ptal-N*01:01 as a structural template) to obtain the model and coordinates. After model building in COOT[59] and several rounds of refinement in Refmac5[60], further refinement was conducted using phenix refine[61]. Subsequently, the MolProbity[62] tools were utilized in Phenix for assessing model quality[63] (refer to Structure data

table). Structure-correlation diagram analysis and plotting were performed using PyMOL[64] (Schrödinger, LLC).

### Structure prediction using AlphaFold
Using 131 sequences of bat MHC class I molecules (with a signal peptide and transmembrane region removed) and bat β2m molecules obtained by downloading from the NCBI database, protein-protein interaction prediction was performed with the help of AlphaFold2[42] (https://colab.research.google.com/github/sokrypton/ColabFold/blob/main/AlphaFold2.ipynb) and AlphaFold-Multimer[43] (https://github.com/deepmind/alphafold) online server, AlphaFold-Multimer uses a weighted combination of pTM and ipTM as the confidence indicator of the models, and finally selecting the model with the highest pLDDT value was used as our template for structure comparison. The prediction structure confidence level is specified in the following equation:

$$model\ confidence = 0.8ipTM + 0.2pTM$$

### Statistics and Reproducibility
The relevant experiments covered in this manuscript are not statistically relevant, and the data obtained regarding the in vitro refolding experiments were repeated more than three times, with consistent results obtained each time.

### Reporting summary
Further information on research design is available in the Nature Portfolio Reporting Summary linked to this article.

### Data availability
The coordinate and structure factors presented in this article have been submitted to the Protein Data Bank[65] (https://www.rcsb.org/) under accession numbers 8HSM,8HSO,8HSW,8HT1, and 8HT9. The mass spectrometry proteomics data have been deposited to the ProteomeXchange Consortium via the PRIDE[66] (https://www.ebi.ac.uk/pride/) partner repository with the dataset identifier PXD047090,PXD047017,and PXD046952. All other data are available from the corresponding author on reasonable request.

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

## Acknowledgements
We thank the staff at the Shanghai Synchrotron Radiation Facility of China for their technical assistance during data collection. This work was supported financially by Major Science and Technology Project of Liaoning Province (2020JH1/10200003). Beijing Science and Technology Project (Z221100006422009). The 2115 Talent Development Program of China Agricultural University.

## Author contributions
N.Z. designed the experiments. S.W. completed all experiments. Z.Q. and X.W. completed part of the crystal structure analysis. L.Z. and S.W predicted the structure of bat MHC-I. L.D. purified the protein. S.W. and N.Z. wrote the manuscript. Funding was provided by N.Z. S.W. and N.Z. analyzed the data and performed statistical analysis.

## Competing interests
The authors declare no competing interests.

## Inclusion and Ethics
Our research complies with the "Inclusion and Ethics" Statement.
