## [Peer Review File · Communications Biology]

Reviewers' comments:

Reviewer #1 (Remarks to the Author):

Major concerns

1. How to explain the bats with MHC I of no 3AA or 5AA insertion. Do they have a more excessive pMHC-I dissociation during flight?
2. The authors proposed potential link between the insertions and adaptation to high body temperatures in flight. But the authors may be careful to conclude this. The artificially-deletion of the serial residues within any of the natural proteins will alleviate the stability of the protein, in the given temperatures, no matter 37 or 41. Actually, the artificially-insertion can also decrease the thermal stability of natural proteins as investigated by the authors in this study. So, it may be untenable for the logic from the instability of mutant Mylu-B*67:01ΔMQQPW to the immunological importance of MQQPW. However, this is still a hypothesis if there is no functional evidence, i.e. the tests of MHC I expression level among the bats during flight, the sustained but moderate T cell responses during the highly-pathogenic virus infection, etc.
3. Although W58 forms a hydrogen bond with W52 in 5AA MHC I and L55 formed a hydrogen bond with W52 in 3AA, the authors may need to indicate whether these interactions are additional compared to OAA MHC I. The MHC I without any insertion may also have some intra-chain interactions in this region.

Minor points

1. COVID-19 has resulted in immense global devastation and loss, with its origins traced back to a bat-borne coronavirus (1, 2). Bats harbor a multitude of viruses, including rabies (3), Hendra virus (HEV) (4, 5), Nipah virus (6, 7), severe acute respiratory syndrome-coronavirus (CoV)
2. Line 75, Line 140, Line 881 "PtaI-N01:01" should be "PtaI-N*01:01".
3. Line 96 41°C of the mixed pMHC-I s complexed with random peptides, demonstrating that both AA and 5AA insertions. AA may be 3AA?
4. Line 256, "However, the pairing of D59 and R65 accounts for only 50% of the 3AA insertion sequences." Over 50% of all available MHC I sequences of bats is not a low proportion. Why the authors used "only".
5. Line 258, Sequence analysis indicates that the effect of the inserted AAs on bat MHC-I molecule does not necessarily rely on the subsequent occurrence of charged residue pairing. Where is the analysis results?
6. Line 281, Using PtaI-N*01:01 as a structural template, we successfully resolved the crystal structure of the pMylu-B*67:01-P1 complex through the molecular replacement method, achieving an impressive resolution of 2.2 Å. Descriptions like this may be moved to the Methods part. Furthermore, 2.2 Å should be 2.2 Å.
7. Figure 1d, the high proportion of D/E in P1 position should be mentioned.
8. Line 290, The presence of the R67 side chain induces the closure of the A pocket, thereby obstructing the exposure of the P1-F side chain (Figure 2. c). It is hard to observe and emphasize the R67 among so many residues in alpha1 helix. The author may want to explain both the function of Y176 and/or R67 in the Fig 2b but failed to mark R67 in this panel.
9. Line 297 "The residue Y7 is conserved across MHC-I molecules", Please refer to some reference or figure to visualize and concretize the description. "while other constituent residues hindered F/Y72 do

not play a crucial role in determining the B pocket.” What does the “other constituent residues hindered F/Y72” mean? The polymorphic residues in position 7 of MHC I or other position or the P2 residue of peptides?

10. Line 299 To address the characters of B and F pockets of Mylu-B67:01, the author used MHC Is from different mammals. However, it may be proper to use similar species for B and F pockets. It seems wired to use Mamu-A01 as the control.

11. Line303 “pHLA-B*:02” should be “pHLA-B*14:02”.

12. Line 325 (Figure 3. c) shall be (Figure 3. e).

13. Line 326 “both wild-type and mutant COVID-19 P1 peptides exhibited better binding to Mylu-B*67:01 (Figure 3. e).” Better than which molecule? “The P1-T mutant peptide displayed the most substantial decrease in complex formation, while the P1-E mutant peptide resulted in the greatest reduction in thermal stability of the complex.” What does this data mean? Does the increasing stability of P1-E mutant with WT Mylu-B has a connection with a forming of hydrogen bond network?

14. Line 337, “potentially forming salt bridges.” Why the word “potentially” was used. The structures in the study can show this answer clearly.

15. Line 338 If “salt bridge network is only formed in the presence of P1-E”, what is the explanation of the decreased binding of P1-F and P1T in Delta5AA mutant?

16. Line 353 “Therefore, there must be another underlying mechanism by which the insertion enhances the peptide binding ability of bat MHC-I.” What is another?

17. Line 359 “(Figure 2. e)” should be “(Figure3. e)”.

18. All the citations of Figure 6. c to Figure 6. in the text are incorrect.

19. Line 372 If “the MQQPW insertion is ... the most unstable part”, why “the insertion of MQQPW reduced the B-factor of the 310 helix”, and had “an improvement in the structural stability of this region”?

20. The authors did not mention whether they used Ptal-N*01:01 and the structures they determined in this study for the deep learning in the structure prediction using AlphaFold.

21. Line 479 The authors may be cautious for the conclusion that 5AA insertion has a stronger enhancement on the thermal stability of pMHC-I. The only evidence is the instability of mutant Mylu-B*67:01ΔMQQPW, but this is artificial results. As the results from this study, artificially-insertion of MQQPW also decreases the thermal stability of natural proteins such as Mylu-A*57.

22. Line 496 “it only involves the insertion itself and applies to all binding peptides”. The author may mean ““it involves not only the insertion itself and can apply to all binding peptides”.

23. Line 751 There is a "1.," before the author's name.

24. Figure 1D. If the authors perform the combined statistical analysis of all the acidic/alkaline pairs of residues in MHC I, it may can reflect their dominant function profile.

25. Figure.8 There is a text box in the bottom left corner of the picture, please delete.

26. Table1 “a=760265” from unit cell parameters of MYLU-B-67ΔMQQPW/P1-F may be a error, please double check.

Reviewer #2 (Remarks to the Author):

The manuscript by Wang and colleagues describes an analysis of peptide binding to a bat MHC-I molecule Mylu-B*67:01 that contains a 5AA insertion in the heavy chain situated at the junction of a 310

helix. The authors use a refolding assay using random peptide libraries (RPLD-MS) to assess peptide binding by integrating bound peptides using de novo sequencing of MHC-I complexes refolded in the presence of the library. This information is combined with assessment of complex thermostability, structural studies of selected complexes along with alpha fold 2 predictions to understand how the insertion of both 3 or 5 amino acids impacts complex stability and peptide ligand selection. Whilst the nature of the data is of interest to the field several aspects of the manuscript need to be explained or reconsidered.

(i) I felt the link between flight, thermal tolerance and the evolution of the MHC-I in bats was significantly oversold and at best remains speculative. In several parts of the manuscript this is stated as fact with little reference to the literature. This emphasis should be removed and stated as speculation in the discussion or conclusion.

(ii) It is not clear which peptides were used to determine peptide binding motifs – what score threshold from the peaks analysis was used to determine this (presumably the score given is average local confidence?) – many of the listed peptides fall below what would generally be considered a strong candidate (ALC>85%). Some of the motifs are not particularly convincing and it is not clear if this is due to promiscuous binding or too few confidently identified binders?

(iii) Given the close relationship of RPLD-MS to more traditional immunopeptidomics analysis the authors should deposit their data and conform to community standards (MIAIPE - Lill JR, et al. Minimal Information About an Immuno-Peptidomics Experiment (MIAIPE). *Proteomics*. 2018; 18(12):e1800110. doi: 10.1002/pmic.201800110).

(iv) Notably thermal melt curves do not show particularly stable MHC-I complexes c.f. human studies for instance where stabilities or Tm values >50C are often observed? Can the authors comment on the measure Tm values relative to other studies using similar assays?

(v) I am not expert in structural prediction made using AF – but it should be made clear these are not necessarily accurate where subtle insertions or mutations are made.

Minor comments

- Line 283 – presumably this is 2.2Å?

- The size exclusion chromatograms of refolded complexes clearly used different columns with different elution volumes - should note expected elution volume of refolded MHC-I for readers reference

(vi) A number of high confidence peptides are reported containing Cysteine residues (which are not supposed to be present in the random peptide libraries used to refold the MHC-I molecules) - if possible the authors should respond to this problem as well.

Reviewer #3 (Remarks to the Author):

In their manuscript, Wang and colleagues present evidence for structural changes on bat MHC molecules upon a 3- or 5-amino acid insertion. These insertions are proposed to stabilize the molecules, and they are part of the bat immune system.

In my review I will focus mainly on the structural work of this manuscript, because this is where my competences are.

The authors present several crystal structures. Unfortunately, they did not insert the main piece of evidence for the structural changes associated with the insertions. In my opinion it is imperative that the authors show an unbiased (2Fo-Fc)-electron density map of the region of interest. The authors should also comment on the B-factors of the atoms of this region. Are they higher, lower or at par with the B-factors of the other atoms of the structures. Only with this evidence it is possible to judge the claims of the authors.

Minor comment here: the authors state an "impressive resolution of 22 Å". First of all, this is like a typo. 22 Å should be 2.2 Å. Second, the authors should refrain from using words such as "impressive". What is impressive for somebody, may not necessarily be impressive for the other. Considering that the average resolution of the structures in the PDB is about 2.0 Å, 2.2 Å does not sound as impressive anymore.

Anyways, the authors should also comment and explain the term B-factors in putty style, which is used in several figures. This is not familiar to everybody.

Finally, the authors should explain what kind of map corrections they did in Coot. In my view, Coot is a visualization program, it does not do any map corrections. Please be concise there.

Another aspect to criticize is the use of AlphaFold2 predictions. These predictions should always be taken with a grain of salt. Of particular importance in this respect is the question, what is the confidence level in particular of the insertion part of the structures. Is this higher, lower or at par with the confidence level of the rest of the structure.

Since much of the manuscript is centered on the question of how the structural alterations bring about thermostability, these issues should be taken seriously by the authors. A figure there would certainly help.

Overall, I find the manuscript a bit too lengthy for the content. Maybe the authors can think of moving some of the present figures to Supplementary Material. Electron density and AlphaFold confidence level should go to the main text.

Dear Editors and Reviewers:

Thank you for your letter and for the reviewers' comments concerning our manuscript entitled "*Two birds with one stone: amino acids insertion in bat MHC-I facilitates maintenance of immune tolerance while enhancing antiviral immunity*" (ID: COMMSBIO-23-2849-T). Those comments are all valuable and very helpful for revising and improving our paper, as well as the important guiding significance to our research. We have studied comments carefully and have made corrections which we hope meet with approval.

Responds to the reviewer's comments:

Reviewer #1 (Remarks to the Author):

Major concerns

1. How to explain the bats with MHC I of no 3AA or 5AA insertion. Do they have a more excessive pMHC-I dissociation during flight?

Response: This is an interesting question. First, we suggest that under evolutionary selection pressure, bat MHC-I without insertion may be inherently more stable, e.g. pMyLu-B*57:01_{RP9} has a higher T_m value than pMyLu-B*67:01ΔMQQPW_{RP9} (new Figure. 5a, h). Thus in flight, these insertion-free pMHC-I complexes may not necessarily undergo excessive degradation. Second, even if these bat pMHC-I complexes without insertion degrade more drastically in flight, it may have little effect on the total amount of pMHC-I complexes, because the proportion of bat MHC-I without insertions is smaller than inserted alleles, at about 18% (new Figure. 6c and Figure S1).

Of course, our explanations for this problem are all speculations based on existing research. More data from cellular or animal experiments would be needed to truly solve this problem, but we are currently unable to perform these experiments.

2. The authors proposed potential link between the insertions and adaptation to high body temperatures in flight. But the authors may be careful to conclude this. The artificially-deletion of the serial residues within any of the natural proteins will alleviate the stability of the protein, in the given temperatures, no matter 37 or 41. Actually, the artificially-insertion can also decrease the thermal stability of natural proteins as investigated by the authors in this study. So, it may be untenable for the logic from the instability of mutant MyLu-B*67:01ΔMQQPW to the immunological importance of MQQPW. However, this is still a hypothesis if there is no functional evidence, i.e. the tests of MHC I expression level among the bats during flight, the sustained but moderate T cell responses during the highly-pathogenic virus infection,

etc.

Response: Thank you for your comments. We associate the insertions with adaptation to high body temperatures for three reasons. First, the immune system of bats does exhibit unique features due to adaptation to flight and high body temperatures (Ref 22 in the manuscript), and the insertion in MHC-I, especially the 5AA insertion, is one of the unique features of the bat immune system. Second, the insertion is right in the 3₁₀ helix region, which is one of the most unstable regions of the MHC-I molecule (Ref 33 and 34). Third, deletion of the natural insertion did reduce the thermal stability of the pMHC-I molecule, making it more susceptible to degradation at 37°C and 41°C. The results of the artificial insertion, on the other hand, suggest that there is fitness between the insertor and the inserted bat MHC-I allele, possibly as a result of evolutionary screening.

Of course, the problems you point out do exist. In the absence of sufficient cellular and in vivo experiments, our over-expression may have allowed some hypotheses to be misinterpreted as conclusions. In the new manuscript, we have made a lot of revisions to weaken the previous over-expression that could easily cause misunderstanding, and clarify the boundary between experimental results and hypotheses.

3. Although W58 forms a hydrogen bond with W52 in 5AA MHC I and L55 formed a hydrogen bond with W52 in 3AA, the authors may need to indicate whether these interactions are additional compared to 0AA MHC I. The MHC I without any insertion may also have some intra-chain interactions in this region.

Response: The bat MHC-I crystal structures now resolved are Mylu-B*67:01, Ptal-N*01:01, and their mutants. If the mutants are used to represent the state of the 0AA insertion, then as noted in the text, no hydrogen bonding of W52 to other residues is indeed found. We also compared the structures of other species of MHC-I molecules, and although some require the involvement of water molecules, hydrogen bonding to W52 is present in some of the structures. This confirms your idea.

We focused on W52 because it plays a key role in peptide assembly and dissociation (Ref 34). If we consider the entire 3₁₀ helix region, intra-chain interactions also exist for 0AA MHC I, but the insertions of 3AA and 5AA bring more interacting forces. Because of this, the overall stability of this region is improved. Your comments made us realize that a comparative analysis of the entire 3₁₀ helix region is more convincing, so we have modified the correlation diagrams (new Figure. 4c, f) and the description of the results (from line 230).

Minor points

1. COVID-19 has resulted in immense global devastation and loss, with its origins traced back to a 44 bat-borne coronavirus (1, 2). Bats harbor a multitude of viruses, including rabies (3), Hendra45 virus (HEV) (4, 5), Nipah virus (6, 7), severe acute respiratory syndrome-coronavirus (CoV)

Response: We have removed the abbreviation (CoV) and fixed the writing errors here (line 44).

2. Line 75, Line140, Line 881 “Ptal-N01:01” should be “Ptal-N*01:01”.

Response: Thank you for the correction, we have made the changes.

3. Line 96 41°C of the mixed pMHC-Is complexed with random peptides, demonstrating that both AA and 5AA insertions. AA may be 3AA?

Response: Thank you for the correction. We have revised the content here based on the review comments and corrected the error (line 91).

4. Line 256, “However, the pairing of D59 and R65 accounts for only 50% of the 3AA insertion sequences.” Over 50% of all available MHC I sequences of bats is not a low proportion. Why the authors used “only”.

Response: Our intention was to point out that this pair of amino acids does not appear in all the 3AA insertion sequences. In response to comments from reviewers and editor, we have made significant changes to the new manuscript. The sentence has been deleted and rephrased in the subsequent results (line 212).

5. Line 258, Sequence analysis indicates that the effect of the inserted AAs on bat MHC-I molecule does not necessarily rely on the subsequent occurrence of charged residue pairing. Where is the analysis results?

Response: Similar to the previous question, in the new manuscript we removed this sentence and made a new statement later (line 212). The statistics of the sequence analysis are showed in the new Figure. S1 and Figure. 6c.

6. Line 281, Using Ptal-N*01:01 as a structural template, we successfully resolved the crystal structure of the pMyly-B*67:01-P1 complex through the molecular replacement method, achieving an impressive resolution of 22 Å. Descriptions like this may be moved to the Methods part. Furthermore, 22 Å should be 2.2 Å.

Response: Thank you for your comments, we made changes and corrected the error (line 142).

7. Figure 1d, the high proportion of D/E in P1 position should be mentioned.

Response: As you suggested, we described this result (line 136).

8. Line 290, The presence of the R67 side chain induces the closure of the A pocket, thereby obstructing the exposure of the P1-F side chain (Figure 2. c). It is hard to observe and emphasize the R67 among so many residues in alpha1 helix. The author may want to explain both the function of Y176 and/or R67 in the Fig 2b but failed to mark R67 in this panel.

Response: We have modified this figure as you commented. We used stick mode to highlight D61 and R67, which is now the new Figure. 1e.

9. Line 297 “The residue Y7 is conserved across MHC-I molecules”, Please refer to some reference or figure to visualize and concretize the description. “while other constituent residues hindered F/Y72 do not play a crucial role in determining the B pocket.” What does the “other constituent residues hindered F/Y72” mean? The polymorphic residues in position 7 of MHC I or other position or the P2 residue of peptides?

Response: Thank you for pointing out the problem. Our previous study (Ref 36) found that Y7 is highly conserved among MHC-I molecules of different species, and therefore does not affect the restriction of the B pocket.

"other constituent residues hindered F/Y72" should be "other constituent residues hindered by F/Y72". In the new manuscript, these words were changed to "Y7 is highly conserved in the vast majority of MHC-I molecules³⁶, and the other constituent residues are blocked by F/Y72 from contacting P2 residue, so none of them are determinants of B pocket preference." (line 158).

10. Line 299 To address the characters of B and F pockets of Mylu-B67:01, the author used MHC Is from different mammals. However, it may be proper to use similar species for B and F pockets. It seems wired to use Mamu-A01 as the control.

Response: We were looking for the B and F pockets that are most similar to Mylu-B67:01 by comparing all solved pMHC-I structures. the F pocket of Mamu-A01 is the most similar to that of Mylu-B67:01, with only a single residue difference (I100L) and very similar bound P9 residues, and therefore was chosen as the control. Of course, if there is a pMHC-I structure in the same or similar species that meets the criteria, it would be a more appropriate reference.

11. Line303 “pHLA-B*:02” should be “pHLA-B*14:02”.

Response: Thank you for the correction, we have made the change (line 166).

12. Line 325 (Figure 3. c) shall be (Figure 3. e).

Response: Thanks for the correction. We have redrawn the Figure and the old Figure. 3e is now the new Figure. 2d (line 185).

13. Line 326 “both wild-type and mutant COVID-19 P1 peptides exhibited better binding to Mylu-B*67:01 (Figure 3. e).” Better than which molecule? “The P1-T mutant peptide displayed the most substantial decrease in complex formation, while the P1-E mutant peptide resulted in the greatest reduction in thermal stability of the complex.” What does this data mean? Does the increasing stability of P1-E mutant with WT Mylu-B has a connection with a forming of hydrogen bond network?

Response: Thanks for the correction. “than Mylu-B*67:01 Δ MQQPW” (line 187).

We agree with your point. The P1-E mutant is able to form a hydrogen bonding network with Mylu-B*67:01, but not with Mylu-B*67:01 Δ MQQPW, and thus the stability of pMylu-B*67:01 Δ MQQPW-P1-E is drastically reduced. This may also be the reason why crystals of the pMylu-B*67:01 Δ MQQPW-P1-E complex could not be obtained (line 189).

14. Line 337, “potentially forming salt bridges.” Why the word “potentially” was used. The structures in the study can show this answer clearly.

Response: Only P1-E allows the formation of hydrogen bonds between D61 and R67, while P1-F and P1-T do not. With reference to your comments, we have revised this part (from line 195) and the new Figure. 3 to make the presentation clearer.

15. Line 338 If “salt bridge network is only formed in the presence of P1-E”, what is the explanation of the decreased binding of P1-F and P1T in Delta5AA mutant?

Response: This is one of the reasons why we believe that D61/R67 is not the only factor. The MQQPW insertion, by enhancing the stabilization of the 3₁₀ helix region, can enhance the stability of the pMylu-B*67:01 complex universally, including P1-F and P1-T. And for the P1-E, this enhancement is also present, but not as pronounced as the enhancement that comes from interactions with D61/R67. We have elaborated on this in the new manuscript (from lines 340 to 366).

16. Line 353 “Therefore, there must be another underlying mechanism by which the insertion enhances the peptide binding ability of bat MHC-I.” What is another?

Response: We have revised this in the new manuscript (line 216). We believe that the

MQQPW insertion strengthens the stabilization of the 3₁₀ helix region as a more general reason.

17. Line 359 “(Figure 2. e)” should be “(Figure 3. e)”.

Response: Thanks for the correction. The old Figure. 3e is now the new Figure. 2d. To make the logic clearer, we have modified the order in which the results are described. This is now located at line 249.

18. All the citations of Figure 6. c to Figure 6. in the text are incorrect.

Response: Thanks for the correction. As mentioned in the previous answer, we adjusted the figures and the results. We first focused on elucidating the improved structural stability and then described its effect on pMHC-I thermal stability and peptide presentation. The old Figure. 5 and Figure. 6 have been newly recombined to become the new Figure 4 and Figure 5. The new Figure 4 and Figure 5 each contain a portion of the old Figure 6.

19. Line 372 If “the MQQPW insertion is ... the most unstable part”, why “the insertion of MQQPW reduced the B-factor of the 3₁₀ helix”, and had “an improvement in the structural stability of this region”?

Response: We believed that MQQPW insertion alters the structure of this region and forms more new hydrogen bonds to maintain the structure, thus reducing the B factors in this region. This part is explained in the new results section "Insertion can enhance the stability of bat pMHC-I by strengthening the 3₁₀ helix region" and new Figure 4.

20. The authors did not mention whether they used Ptal-N*01:01 and the structures they determined in this study for the deep learning in the structure prediction using AlphaFold.

Response: We did not train AF2 with data of Ptal-N*01:01 and Mylu-B*67:01. This is explained in the discussion section of our new manuscript (line 392).

21. Line 479 The authors may be cautious for the conclusion that 5AA insertion has a stronger enhancement on the thermal stability of pMHC-I. The only evidence is the instability of mutant Mylu-B*67:01ΔMQQPW, but this is artificial results. As the results from this study, artificially-insertion of MQQPW also decreases the thermal stability of natural proteins such as Mylu-A*57.

Response: Your comment is correct that this conclusion is hasty. In the new manuscript, we only mentioned this possibility at the end of the discussion, and limited it to the

case where the inserted and inserted sequences match, to provide a possible explanation for the unique 5AA insertion in bats.

22. Line 496 “it only involves the insertion itself and applies to all binding peptides”. The author may mean ““it involves not only the insertion itself and can apply to all binding peptides””.

Response: Thanks, we have made the correction (line 359).

23. Line 751 There is a "1.," before the author's name.

Response: Thanks, we have made the correction.

24. Figure 1D. If the authors perform the combined statistical analysis of all the acidic/alkaline pairs of residues in MHC I, it may can reflect their dominant function profile.

Response: In the new manuscript, these statistics were placed in the new Figure.S1c and mentioned in the results (Line 213).

25. Figure.8 There is a text box in the bottom left corner of the picture, please delete.

Response: Thanks, we have made the correction.

26. Table1 “a=760265” from unit cell parameters of MYLU-B-67ΔMQQPW/P1-F may be a error, please double check.

Response: Thanks, it should be “a=76.0265”, we have made the correction.

Special thanks to you for your good comments.

Reviewer #2 (Remarks to the Author):

The manuscript by Wang and colleagues describes an analysis of peptide binding to a bat MHC-I molecule Mylu-B*67:01 that contains a 5AA insertion in the heavy chain situated at the junction of a 3_{10} helix. The authors use a refolding assay using random peptide libraries (RPLD-MS) to assess peptide binding by integrating bound peptides using de novo sequencing of MHC-I complexes refolded in the presence of the library. This information is combined with assessment of complex thermostability, structural studies of selected complexes along with alpha fold 2 predictions to understand how the insertion of both 3 or 5 amino acids impacts complex stability and peptide ligand selection. Whilst the nature of the data is of interest to the field several aspects of the manuscript need to be explained or reconsidered.

(i) I felt the link between flight, thermal tolerance and the evolution of the MHC-I in bats was significantly oversold and at best remains speculative. In several parts of the manuscript this is stated as fact with little reference to the literature. This emphasis should be removed and stated as speculation in the discussion or conclusion.

Response: Thank you for your comments. Other reviewers also raised similar questions. Without sufficient cellular and in vivo experiments, our previous presentation was over-expressed, which might make some hypotheses mistaken as conclusions. In the newly submitted version, we have made a lot of revisions, weakened the previous over-expression which is easy to cause misunderstanding, and clarified the boundaries between experimental results and hypotheses.

(ii) It is not clear which peptides were used to determine peptide binding motifs – what score threshold from the peaks analysis was used to determine this (presumably the score given is average local confidence?) – many of the listed peptides fall below what would generally be considered a strong candidate (ALC>85%). Some of the motifs are not particularly convincing and it is not clear if this is due to promiscuous binding or too few confidently identified binders?

Response: We appreciate your interest in RPLD-MS technology. De novo MS technology allows us to sequence peptides in random peptide libraries that bind to target MHC-I, which is the basis for the established RPLD-MS technology. De novo analysis of MS is a non-data-dependent peptide MS sequencing technology. It does not need a specific target protein database to match the peptide sequences but determines the sequences directly by a specific algorithm according to the MS signals. The algorithm will score the matching degree between the peptide sequence and the corresponding MS signal. High scores represent high reliability.

We generally perform integrated analysis on peptides with scores ≥ 50 . The results of RPLD-MS were reproducible. At present, we have determined the peptide binding motifs of MHC-I molecules of different species by RPLD-MS technology, and the accuracy of the results has been verified by in vitro renaturation binding experiments and crystal structures (bat, PMID: 31076531; shark, PMID: 34145057; frog, PMID: 31776204; lizard, PMID: 33637616; pig, PMID: 33717070; duck, PMID: 35623661).

The PBM of Mylu-B*67:01 Δ MQQPW is not as clear as Mylu-B*67:01. We believe this is due to a large reduction in bound peptides (new Figure 2. A).

(iii) Given the close relationship of RPLD-MS to more traditional immunopeptidomics analysis the authors should deposit their data and conform to

community standards (MIAIPE - Lill JR, et al. Minimal Information About an Immuno-Peptidomics Experiment (MIAIPE). *Proteomics*. 2018; 18(12):e1800110. doi: 10.1002/pmic.201800110).

Response: Thank you for your suggestion, we have uploaded the relevant data and described it in the Methods (line 514).

(iv) Notably thermal melt curves do not show particularly stable MHC-I complexes c.f. human studies for instance where stabilities or T_m values >50C are often observed? Can the authors comment on the measure T_m values relative to other studies using similar assays?

Response: This is a very sharp question. In fact, we have found this problem in other animal MHC-I studies and have been asked about it by other reviewers. We believed that this is mainly because our peptides were screened by in vitro methods and have a high probability of not being able to tolerate body temperature for a long time and remain stable (new Figure. 5b and e). In contrast, most of the HLA-I binding peptides were captured from cells or in vivo, which can tolerate body temperature and multiple washes during purification without dissociation. These are higher affinity binding peptides and therefore tend to have higher T_m values.

At present, the screening of animal MHC-I restricted T cell epitopes is often limited by experimental conditions, such as the lack of high-affinity MHC-I monoclonal antibodies, cell lines, and experimental animals with clear haplotype. We don't have those conditions to study bat MHC-I now and need to do more work to improve it gradually.

(v) I am not an expert in structural prediction made using AF – but it should be made clear these are not necessarily accurate where subtle insertions or mutations are made.

Response: We strongly agree with you that assessing the credibility of AF2 predictions for polymorphic molecules and their complexes requires caution. In this study, we compared the predictions with real crystal structures and confirmed the correctness of the AF2 predictions. This suggested that AF2 has the ability to correctly predict the structure of bat MHC-I.

Of course, we could not guarantee the correctness of all predicted bat MHC-I structures. However, sequence comparison revealed that these insertions are still somewhat conserved, and the predicted results reflected this similarity (new Figure. 6c). It is also worth mentioning that there are some studies applying AF2 to HLA structure prediction and epitope screening with good results (Ref 44 and 45 in Discuss). Therefore, we believe that the AF2 predictions in this study can be used.

Minor comments

- Line 283 – presumably this is 2.2Å?

Response: Thanks, we have made the correction (line 142).

- The size exclusion chromatograms of refolded complexes clearly used different columns with different elution volumes - should note expected elution volume of refolded MHC-I for readers reference

Response: Thanks to your comment, we have added a note into the Figure legend (new Figure. 1a).

- (vi) A number of high confidence peptides are reported containing Cysteine residues (which are not supposed to be present in the random peptide libraries used to refold the MHC-I molecules) - if possible the authors should respond to this problem as well.

Response: Thank you for your careful review. This was also questioned by a Reviewer in a previous study. This problem is actually caused by De Novo MS result analysis, the non-data-dependent MS identification approach, just like mentioned in your question (ii).

In previous studies, we validated a number of peptides with high scores but clearly not conforming to the peptide binding motif, and none of them could bind. This suggested that the specific peptides obtained by De Novo MS sequencing should not be credible. When identifying specific peptide sequences, the De Novo algorithm has the possibility of misidentifying certain signals as cysteines. However, in terms of overall statistics, the results produced by the De Novo MS analysis are credible. There may be certain errors in a specific peptide in the random peptide library generated by RPLD-MS, but the overall characteristics of all random peptides do conform to the peptide binding motif of the tested MHC-I molecule, whether from the perspective of structure or in vitro refolding verification. From the overall statistics, cysteine appeared at a very low percentage and did not have a significant effect on the PBM logo.

Thanks again to the reviewers for their valuable suggestions on our article.

Reviewer #3 (Remarks to the Author):

In their manuscript, Wang and colleagues present evidence for structural changes on bat MHC molecules upon a 3- or 5-amino acid insertion. These insertions are proposed to stabilize the molecules, and they are part of the bat immune system.

In my review I will focus mainly on the structural work of this manuscript, because

this is where my competences are.

The authors present several crystal structures. Unfortunately, they did not insert the main piece of evidence for the structural changes associated with the insertions. In my opinion it is imperative that the authors show an unbiased (2Fo-Fc)-electron density map of the region of interest. The authors should also comment on the B-factors of the atoms of this region. Are they higher, lower or at par with the B-factors of the other atoms of the structures. Only with this evidence it is possible to judge the claims of the authors.

Response: Thank you for your comments. We added the unbiased (2Fo-Fc)-electron density maps to the new figures 1, 2, and 4. We also added a B factor comparison in the Results (new Figure. 4a and line 223).

Minor comment here: the authors state an "impressive resolution of 22 Å". First of all, this is like a typo. 22 Å should be 2.2 Å. Second, the authors should refrain from using words such as "impressive". What is impressive for somebody, may not necessarily be impressive for the other. Considering that the average resolution of the structures in the PDB is about 2.0 Å, 2.2 Å does not sound as impressive anymore.

Response: Thanks, we have corrected (line 142).

Anyways, the authors should also comment and explain the term B-factors in putty style, which is used in several figures. This is not familiar to everybody.

Response: We provided an introduction to the B-factor putty model (line 227). In addition, as mentioned before, we also provided more comparisons of B factors in the region of interest (new Figure 4A and line 223).

Finally, the authors should explain what kind of map corrections they did in Coot. In my view, Coot is a visualization program, it does not do any map corrections. Please be concise there.

Response: Coot is used to manually build the structural model based on the (2Fo-Fc)-electron density map. We have corrected, thanks for your comment.

Another aspect to criticize is the use of Alphafold2 predictions. These predictions should always be taken with a grain of salt. Of particular importance in this respect is the question, what is the confidence level in particular of the insertion part of the structures. Is this higher, lower or at par with the confidence level of the rest of the structure.

Response: Your comments are important. We added the presentation and description of the confidence of the predicted results (Figure. 6a and line 289). The confidence in the predicted Mylu-B*67:01 heavy chain structure predicted by AF2 is high (pLDDT>90), including the small helix formed by the MQQPW insertion. The overall confidence in the predicted complex structure is also high, with relatively low confidence in the binding peptide and the loop at Δ MQQPW ($90 > \text{pLDDT} > 70$). Only the exposing residues of Mylu-B*67:01 Δ MQQPW bound peptide are with low confidence. These results suggested that AF2 was relatively accurate in its predictions of α -helices and β -sheets, while the confidence in its predictions of more flexible rings was relatively low.

Although the AF2 predictions cannot be equated to the resolved structures, we believe that the predictions for bat MHC-Is can be used in this study. In addition to the high confidence of the prediction results, sequence comparison revealed that these insertions are somewhat conserved, and the predicted results reflected this similarity (new Figure. 6c). In addition, there have been some studies applying AF2 to HLA structure prediction and epitope screening with good results (Discussion inside 44,45).

Since much of the manuscript is centered on the question of how the structural alterations bring about thermostability, these issues should be taken seriously by the authors. A figure there would certainly help.

Response: To make the logic clearer, we have modified the order in which the results are described. We first focused on elucidating the improved structural stability and then described its effect on pMHC-I thermal stability and peptide presentation. The old Figure. 5 and Figure. 6 have been newly recombined to become the new Figure 4 and Figure 5. This provides a clear illustration of how increased structural stability affects the thermostability of pMHC-I.

Overall, I find the manuscript a bit too lengthy for the content. Maybe the authors can think of moving some of the present figures to Supplementary Material. Electron density and Alphafold confidence level should go to the main text.

Response: We reduced a figure in the main text based on your comments. Part of it was put into Supplementary Material, while the electron cloud density and AF2 confidence were put into the main text.

We sincerely thank the reviewers for their suggestions to us regarding the crystal structure.

We appreciate for Editors/Reviewers' warm work earnestly, and hope that the

correction will meet with approval.

Once again, thank you very much for your comments and suggestions.

Reviewers' comments:

Reviewer #1 (Remarks to the Author):

The authors have addressed all my concerns.

Reviewer #3 (Remarks to the Author):

In their revised manuscript, the authors have taken my comments into account, albeit in an insufficient manner.

When I read the rebuttal letter, I first thought everything was fine, but what I found in the manuscript made me change my mind again.

Most importantly, I asked for an unbiased 2Fo-Fc electron density map as evidence for the conformation of the insertion. In figure 1d, the electron density map is not at all described in the figure legend, and in figures 2 and 4 there is no mention of the word "unbiased". What I mean is that the structure needs to be refined without the amino acids in question present, and based on this phase set an electron density map needs to be calculated which shows the structure of the insertion (or not). It also needs to be mentioned at which contour level the electron density map is displayed. Since the electron density maps in Figure 1, 2 and 4 are rather hard to see, it may be beneficial to add the correlation coefficient between map and model for the parts of the structure in question.

Small comment on the side: it is not "electronic density map" but "electron density map" and it is also not "electron cloud density map" or some combination of similar terms. The authors really need to be precise in what they are talking about.

The explanation of the B-factor putty model in the text is also insufficient. What do the colors mean? I understand that the model arises from the comparison of B-factors, but the authors need to be a bit more careful to explain to the reader what one sees in the figures.

Generally, I find the figures way too small and hard to read. I understand that the goal is to get as much information into the figures as possible, but this is not always very helpful to the reader. I wonder how many readers will zoom into to 200% to see what one is supposed to see at normal size.

Another small question: was crystallisation really achieved at protein concentrations of 2-4 micro-g/ml or rather at the more usual concentrations 2-4 mg/ml?

Dear Editors and Reviewers:

Thank you for your letter and for the reviewers' comments concerning our manuscript entitled "*Two birds with one stone: amino acids insertion in bat MHC-I facilitates maintenance of immune tolerance while enhancing antiviral immunity*" (ID: COMMSBIO-23-2849-T). Those comments are all valuable and very helpful for revising and improving our paper, as well as the important guiding significance to our research. We have studied comments carefully and have made corrections which we hope meet with approval.

Updated responses:

Reviewer #3 :

In their revised manuscript, the authors have taken my comments into account, albeit in an insufficient manner.

When I read the rebuttal letter, I first thought everything was fine, but what I found in the manuscript made me change my mind again.

Most importantly, I asked for an unbiased 2Fo-Fc electron density map as evidence for the conformation of the insertion. In figure 1d, the electron density map is not at all described in the figure legend, and in figures 2 and 4 there is no mention of the word "unbiased". What I mean is that the structure needs to be refined without the amino acids in question present, and based on this phase set an electron density map needs to be calculated which shows the structure of the insertion (or not). It also needs to be mentioned at which contour level the electron density map is displayed. Since the electron density maps in Figure 1, 2 and 4 are rather hard to see, it may be beneficial to add the correlation coefficient between map and model for the parts of the structure in question.

Response: Thank you for your comments and suggestions. We have revised the figures you mentioned and rewritten the figure legend. The unbiased 2Fo-Fc electron density maps, which you were particularly interested in, are now displayed in the new Figure 2(f). We removed MQQPW and generated the mtz file using phenix.composite_omit_map, and transformed it into a map file using the CCP4 Package, and the unbiased 2Fo-Fc electron density map was visualized using Pymol. The correlation coefficient values between map and model for the inserted MQQPW of the three pMyly-B*67:01 structures were also added into the new figure legend. The results indicate that the crystal structures we have determined are credible, and the structural differences are indeed caused by the insertion of MQQPW. Regarding the issue of the electron density maps being difficult to observe, we have also made modifications. Previously, we had reduced the mesh_width parameter in

Pymol to make the electron density maps less prominent. However, in the composite figures, they indeed appeared insufficiently clear. Therefore, we have reverted to the default value of mesh_width and created new figures, making the electron density maps appear clearer. Additionally, we have removed the electron density maps from Figure 4, as they obscured the hydrogen bonds and made the figures look cluttered.

Small comment on the side: it is not "electronic density map" but "electron density map" and it is also not "electron cloud density map" or some combination of similar terms. The authors really need to be precise in what they are talking about.

Response: Thank you for your suggestion. We have reviewed these terms and made the necessary corrections.

The explanation of the B-factor putty model in the text is also insufficient. What do the colors mean? I understand that the model arises from the comparison of B-factors, but the authors need to be a bit more careful to explain to the reader what one sees in the figures.

Response: Thank you for your suggestion. We have reviewed these terms and made the necessary corrections. We have added an introduction to the display format of the B-factor putty model in the figure legend, including the color and thickness.

Generally, I find the figures way too small and hard to read. I understand that the goal is to get as much information into the figures as possible, but this is not always very helpful to the reader. I wonder how many readers will zoom into to 200% to see what one is supposed to see at normal size.

Response: We have made modifications to the figures based on your suggestions, as detailed in the response above.

Another small question: was crystallisation really achieved at protein concentrations of 2-4 micro-g/ml or rather at the more usual concentrations 2-4 mg/ml?

Response: The protein concentration we wrote was 2-4 $\mu\text{g}/\mu\text{l}$, which is equivalent to 2-4 mg/ml. We have already corrected it to the more common 2-4 mg/ml (line 542).

Responds to the reviewer's comments:

Reviewer #1 (Remarks to the Author):

Major concerns

1. How to explain the bats with MHC I of no 3AA or 5AA insertion. Do they have a more excessive pMHC-I dissociation during flight?

Response: This is an interesting question. First, we suggest that under evolutionary selection pressure, bat MHC-I without insertion may be inherently more stable, e.g. pMyIu-B*57:01_{RP9} has a higher T_m value than pMyIu-B*67:01ΔMQQPW_{RP9} (new Figure. 5a, h). Thus in flight, these insertion-free pMHC-I complexes may not necessarily undergo excessive degradation. Second, even if these bat pMHC-I complexes without insertion degrade more drastically in flight, it may have little effect on the total amount of pMHC-I complexes, because the proportion of bat MHC-I without insertions is smaller than inserted alleles, at about 18% (new Figure. 6c and Figure S1).

Of course, our explanations for this problem are all speculations based on existing research. More data from cellular or animal experiments would be needed to truly solve this problem, but we are currently unable to perform these experiments.

2. The authors proposed potential link between the insertions and adaptation to high body temperatures in flight. But the authors may be careful to conclude this. The artificially-deletion of the serial residues within any of the natural proteins will alleviate the stability of the protein, in the given temperatures, no matter 37 or 41. Actually, the artificially-insertion can also decrease the thermal stability of natural proteins as investigated by the authors in this study. So, it may be untenable for the logic from the instability of mutant MyIu-B*67:01ΔMQQPW to the immunological importance of MQQPW. However, this is still a hypothesis if there is no functional evidence, i.e. the tests of MHC I expression level among the bats during flight, the sustained but moderate T cell responses during the highly-pathogenic virus infection, etc.

Response: Thank you for your comments. We associate the insertions with adaptation to high body temperatures for three reasons. First, the immune system of bats does exhibit unique features due to adaptation to flight and high body temperatures (Ref 22 in the manuscript), and the insertion in MHC-I, especially the 5AA insertion, is one of the unique features of the bat immune system. Second, the insertion is right in the 3₁₀ helix region, which is one of the most unstable regions of the MHC-I molecule (Ref 33 and 34). Third, deletion of the natural insertion did reduce the thermal stability of the pMHC-I molecule, making it more susceptible to degradation at 37°C and 41°C. The results of the artificial insertion, on the other hand, suggest that there is fitness between the insertor and the inserted bat MHC-I allele, possibly as a result of evolutionary screening.

Of course, the problems you point out do exist. In the absence of sufficient cellular and in vivo experiments, our over-expression may have allowed some hypotheses to

be misinterpreted as conclusions. In the new manuscript, we have made a lot of revisions to weaken the previous over-expression that could easily cause misunderstanding, and clarify the boundary between experimental results and hypotheses.

3. Although W58 forms a hydrogen bond with W52 in 5AA MHC I and L55 formed a hydrogen bond with W52 in 3AA, the authors may need to indicate whether these interactions are additional compared to 0AA MHC I. The MHC I without any insertion may also have some intra-chain interactions in this region.

Response: The bat MHC-I crystal structures now resolved are Mylu-B*67:01, Ptal-N*01:01, and their mutants. If the mutants are used to represent the state of the 0AA insertion, then as noted in the text, no hydrogen bonding of W52 to other residues is indeed found. We also compared the structures of other species of MHC-I molecules, and although some require the involvement of water molecules, hydrogen bonding to W52 is present in some of the structures. This confirms your idea.

We focused on W52 because it plays a key role in peptide assembly and dissociation (Ref 34). If we consider the entire 3_{10} helix region, intra-chain interactions also exist for 0AA MHC I, but the insertions of 3AA and 5AA bring more interacting forces. Because of this, the overall stability of this region is improved. Your comments made us realize that a comparative analysis of the entire 3_{10} helix region is more convincing, so we have modified the correlation diagrams (new Figure. 4c, f) and the description of the results (from line 230) (Updated 234).

Minor points

1. COVID-19 has resulted in immense global devastation and loss, with its origins traced back to a 44 bat-borne coronavirus (1, 2). Bats harbor a multitude of viruses, including rabies (3), Hendra45 virus (HEV) (4, 5), Nipah virus (6, 7), severe acute respiratory syndrome-coronavirus (CoV)

Response: We have removed the abbreviation (CoV) and fixed the writing errors here (line 44) (Updated 44).

2. Line 75, Line140, Line 881 “Ptal-N01:01” should be “Ptal-N*01:01”.

Response: Thank you for the correction, we have made the changes.

3. Line 96 41°C of the mixed pMHC-Is complexed with random peptides, demonstrating that both AA and 5AA insertions. AA may be 3AA?

Response: Thank you for the correction. We have revised the content here based on the review comments and corrected the error (line 91) (Updated 91).

4. Line 256, “However, the pairing of D59 and R65 accounts for only 50% of the 3AA insertion sequences.” Over 50% of all available MHC I sequences of bats is not a low proportion. Why the authors used “only”.

Response: Our intention was to point out that this pair of amino acids does not appear in all the 3AA insertion sequences. In response to comments from reviewers and editor, we have made significant changes to the new manuscript. The sentence has been deleted and rephrased in the subsequent results (line 212) (Updated 216).

5. Line 258, Sequence analysis indicates that the effect of the inserted AAs on bat MHC-I molecule does not necessarily rely on the subsequent occurrence of charged residue pairing. Where is the analysis results?

Response: Similar to the previous question, in the new manuscript we removed this sentence and made a new statement later (line 212) (Updated 216). The statistics of the sequence analysis are showed in the new Figure. S1 and Figure. 6c.

6. Line 281, Using Ptal-N*01:01 as a structural template, we successfully resolved the crystal structure of the pMyly-B*67:01-P1 complex through the molecular replacement method, achieving an impressive resolution of 22 Å. Descriptions like this may be moved to the Methods part. Furthermore, 22 Å should be 2.2 Å.

Response: Thank you for your comments, we made changes and corrected the error (line 142) (Updated 142).

7. Figure 1d, the high proportion of D/E in P1 position should be mentioned.

Response: As you suggested, we described this result (line 136) (Updated 136).

8. Line 290, The presence of the R67 side chain induces the closure of the A pocket, thereby obstructing the exposure of the P1-F side chain (Figure 2. c). It is hard to observe and emphasize the R67 among so many residues in alpha1 helix. The author may want to explain both the function of Y176 and/or R67 in the Fig 2b but failed to mark R67 in this panel.

Response: We have modified this figure as you commented. We used stick mode to highlight D61 and R67, which is now the new Figure. 1e.

9. Line 297 “The residue Y7 is conserved across MHC-I molecules”, Please refer to some reference or figure to visualize and concretize the description. “while other constituent residues hindered F/Y72 do not play a crucial role in determining the B

pocket.” What does the “other constituent residues hindered F/Y72” mean? The polymorphic residues in position 7 of MHC I or other position or the P2 residue of peptides?

Response: Thank you for pointing out the problem. Our previous study (Ref 36) found that Y7 is highly conserved among MHC-I molecules of different species, and therefore does not affect the restriction of the B pocket.

"other constituent residues hindered F/Y72" should be "other constituent residues hindered by F/Y72". In the new manuscript, these words were changed to "Y7 is highly conserved in the vast majority of MHC-I molecules³⁶, and the other constituent residues are blocked by F/Y72 from contacting P2 residue, so none of them are determinants of B pocket preference." (line 158) (Updated 158).

10. Line 299 To address the characters of B and F pockets of Mylu-B67:01, the author used MHC Is from different mammals. However, it may be proper to use similar species for B and F pockets. It seems wired to use Mamu-A01 as the control.

Response: We were looking for the B and F pockets that are most similar to Mylu-B67:01 by comparing all solved pMHC-I structures. the F pocket of Mamu-A01 is the most similar to that of Mylu-B67:01, with only a single residue difference (I100L) and very similar bound P9 residues, and therefore was chosen as the control. Of course, if there is a pMHC-I structure in the same or similar species that meets the criteria, it would be a more appropriate reference.

11. Line303 “pHLA-B*:02” should be “pHLA-B*14:02”.

Response: Thank you for the correction, we have made the change (line 166) (Updated 166).

12. Line 325 (Figure 3. c) shall be (Figure 3. e).

Response: Thanks for the correction. We have redrawn the Figure and the old Figure. 3e is now the new Figure. 2d (line 185) (Updated 185).

13. Line 326 “both wild-type and mutant COVID-19 P1 peptides exhibited better binding to Mylu-B*67:01 (Figure 3. e).” Better than which molecule? “The P1-T mutant peptide displayed the most substantial decrease in complex formation, while the P1-E mutant peptide resulted in the greatest reduction in thermal stability of the complex.” What does this data mean? Does the increasing stability of P1-E mutant with WT Mylu-B has a connection with a forming of hydrogen bond network?

Response: Thanks for the correction. “than Mylu-B*67:01ΔMQQPW” (line 187)

(Updated 187).

We agree with your point. The P1-E mutant is able to form a hydrogen bonding network with Mylu-B*67:01, but not with Mylu-B*67:01ΔMQQPW, and thus the stability of pMylu-B*67:01ΔMQQPW-P1-E is drastically reduced. This may also be the reason why crystals of the pMylu-B*67:01ΔMQQPW-P1-E complex could not be obtained (line 189) (Updated 189).

14. Line 337, “potentially forming salt bridges.” Why the word “potentially” was used. The structures in the study can show this answer clearly.

Response: Only P1-E allows the formation of hydrogen bonds between D61 and R67, while P1-F and P1-T do not. With reference to your comments, we have revised this part (from line 195) (Updated 199) and the new Figure. 3 to make the presentation clearer.

15. Line 338 If “salt bridge network is only formed in the presence of P1-E”, what is the explanation of the decreased binding of P1-F and P1T in Delta5AA mutant?

Response: This is one of the reasons why we believe that D61/R67 is not the only factor. The MQQPW insertion, by enhancing the stabilization of the 3₁₀ helix region, can enhance the stability of the pMylu-B*67:01 complex universally, including P1-F and P1-T. And for the P1-E, this enhancement is also present, but not as pronounced as the enhancement that comes from interactions with D61/R67. We have elaborated on this in the new manuscript (from lines 340 to 366) (Updated 344 to 370).

16. Line 353 “Therefore, there must be another underlying mechanism by which the insertion enhances the peptide binding ability of bat MHC-I.” What is another?

Response: We have revised this in the new manuscript (line 216) (Updated 220). We believe that the MQQPW insertion strengthens the stabilization of the 3₁₀ helix region as a more general reason.

17. Line 359 “(Figure 2. e)” should be “(Figure 3. e)”.

Response: Thanks for the correction. The old Figure. 3e is now the new Figure. 2d. To make the logic clearer, we have modified the order in which the results are described. This is now located at line 249 (Updated 253).

18. All the citations of Figure 6. c to Figure 6. in the text are incorrect.

Response: Thanks for the correction. As mentioned in the previous answer, we adjusted the figures and the results. We first focused on elucidating the improved structural

stability and then described its effect on pMHC-I thermal stability and peptide presentation. The old Figure. 5 and Figure. 6 have been newly recombined to become the new Figure 4 and Figure 5. The new Figure 4 and Figure 5 each contain a portion of the old Figure 6.

19. Line 372 If “the MQQPW insertion is ... the most unstable part”, why “the insertion of MQQPW reduced the B-factor of the 3₁₀ helix”, and had “an improvement in the structural stability of this region”?

Response: We believed that MQQPW insertion alters the structure of this region and forms more new hydrogen bonds to maintain the structure, thus reducing the B factors in this region. This part is explained in the new results section "Insertion can enhance the stability of bat pMHC-I by strengthening the 3₁₀ helix region" and new Figure 4.

20. The authors did not mention whether they used Ptal-N*01:01 and the structures they determined in this study for the deep learning in the structure prediction using AlphaFold.

Response: We did not train AF2 with data of Ptal-N*01:01 and Mylu-B*67:01. This is explained in the discussion section of our new manuscript (line 392) (Updated 396).

21. Line 479 The authors may be cautious for the conclusion that 5AA insertion has a stronger enhancement on the thermal stability of pMHC-I. The only evidence is the instability of mutant Mylu-B*67:01ΔMQQPW, but this is artificial results. As the results from this study, artificially-insertion of MQQPW also decreases the thermal stability of natural proteins such as Mylu-A*57.

Response: Your comment is correct that this conclusion is hasty. In the new manuscript, we only mentioned this possibility at the end of the discussion, and limited it to the case where the inserted and inserted sequences match, to provide a possible explanation for the unique 5AA insertion in bats.

22. Line 496 “it only involves the insertion itself and applies to all binding peptides”. The author may mean ““it involves not only the insertion itself and can apply to all binding peptides”.

Response: Thanks, we have made the correction (line 359) (Updated 363).

23. Line 751 There is a "1.," before the author's name.

Response: Thanks, we have made the correction.

24. Figure 1D. If the authors perform the combined statistical analysis of all the acidic/alkaline pairs of residues in MHC I, it may can reflect their dominant function profile.

Response: In the new manuscript, these statistics were placed in the new Figure.S1c and mentioned in the results (Line 213) (Updated 217).

25. Figure.8 There is a text box in the bottom left corner of the picture, please delete.

Response: Thanks, we have made the correction.

26. Table1 “a=760265” from unit cell parameters of MYLU-B-67ΔMQQPW/P1-F may be a error, please double check.

Response: Thanks, it should be “a=76.0265”, we have made the correction.

Special thanks to you for your good comments.

Reviewer #2 (Remarks to the Author):

The manuscript by Wang and colleagues describes an analysis of peptide binding to a bat MHC-I molecule Mylu-B*67:01 that contains a 5AA insertion in the heavy chain situated at the junction of a 3_{10} helix. The authors use a refolding assay using random peptide libraries (RPLD-MS) to assess peptide binding by integrating bound peptides using de novo sequencing of MHC-I complexes refolded in the presence of the library. This information is combined with assessment of complex thermostability, structural studies of selected complexes along with alpha fold 2 predictions to understand how the insertion of both 3 or 5 amino acids impacts complex stability and peptide ligand selection. Whilst the nature of the data is of interest to the field several aspects of the manuscript need to be explained or reconsidered.

(i) I felt the link between flight, thermal tolerance and the evolution of the MHC-I in bats was significantly oversold and at best remains speculative. In several parts of the manuscript this is stated as fact with little reference to the literature. This emphasis should be removed and stated as speculation in the discussion or conclusion.

Response: Thank you for your comments. Other reviewers also raised similar questions. Without sufficient cellular and in vivo experiments, our previous presentation was over-expressed, which might make some hypotheses mistaken as conclusions. In the newly submitted version, we have made a lot of revisions, weakened the previous over-expression which is easy to cause misunderstanding, and clarified the boundaries between experimental results and hypotheses.

(ii) It is not clear which peptides were used to determine peptide binding motifs – what score threshold from the peaks analysis was used to determine this (presumably the score given is average local confidence?) – many of the listed peptides fall below what would generally be considered a strong candidate (ALC>85%). Some of the motifs are not particularly convincing and it is not clear if this is due to promiscuous binding or too few confidently identified binders?

Response: We appreciate your interest in RPLD-MS technology. De novo MS technology allows us to sequence peptides in random peptide libraries that bind to target MHC-I, which is the basis for the established RPLD-MS technology. De novo analysis of MS is a non-data-dependent peptide MS sequencing technology. It does not need a specific target protein database to match the peptide sequences but determines the sequences directly by a specific algorithm according to the MS signals. The algorithm will score the matching degree between the peptide sequence and the corresponding MS signal. High scores represent high reliability.

We generally perform integrated analysis on peptides with scores ≥ 50 . The results of RPLD-MS were reproducible. At present, we have determined the peptide binding motifs of MHC-I molecules of different species by RPLD-MS technology, and the accuracy of the results has been verified by in vitro renaturation binding experiments and crystal structures (bat, PMID: 31076531; shark, PMID: 34145057; frog, PMID: 31776204; lizard, PMID: 33637616; pig, PMID: 33717070; duck, PMID: 35623661).

The PBM of Mylu-B*67:01 Δ MQQPW is not as clear as Mylu-B*67:01. We believe this is due to a large reduction in bound peptides (new Figure 2. A).

(iii) Given the close relationship of RPLD-MS to more traditional immunopeptidomics analysis the authors should deposit their data and conform to community standards (MIAIPE - Lill JR, et al. Minimal Information About an Immuno-Peptidomics Experiment (MIAIPE). *Proteomics*. 2018; 18(12):e1800110. doi: 10.1002/pmic.201800110).

Response: Thank you for your suggestion, we have uploaded the relevant data and described it in the Methods (line 514) (Updated 518).

(iv) Notably thermal melt curves do not show particularly stable MHC-I complexes c.f. human studies for instance where stabilities or T_m values >50C are often observed? Can the authors comment on the measure T_m values relative to other studies using similar assays?

Response: This is a very sharp question. In fact, we have found this problem in other

animal MHC-I studies and have been asked about it by other reviewers. We believed that this is mainly because our peptides were screened by in vitro methods and have a high probability of not being able to tolerate body temperature for a long time and remain stable (new Figure. 5b and e). In contrast, most of the HLA-I binding peptides were captured from cells or in vivo, which can tolerate body temperature and multiple washes during purification without dissociation. These are higher affinity binding peptides and therefore tend to have higher T_m values.

At present, the screening of animal MHC-I restricted T cell epitopes is often limited by experimental conditions, such as the lack of high-affinity MHC-I monoclonal antibodies, cell lines, and experimental animals with clear haplotype. We don't have those conditions to study bat MHC-I now and need to do more work to improve it gradually.

(v) I am not an expert in structural prediction made using AF – but it should be made clear these are not necessarily accurate where subtle insertions or mutations are made.

Response: We strongly agree with you that assessing the credibility of AF2 predictions for polymorphic molecules and their complexes requires caution. In this study, we compared the predictions with real crystal structures and confirmed the correctness of the AF2 predictions. This suggested that AF2 has the ability to correctly predict the structure of bat MHC-I.

Of course, we could not guarantee the correctness of all predicted bat MHC-I structures. However, sequence comparison revealed that these insertions are still somewhat conserved, and the predicted results reflected this similarity (new Figure. 6c). It is also worth mentioning that there are some studies applying AF2 to HLA structure prediction and epitope screening with good results (Ref 44 and 45 in Discuss). Therefore, we believe that the AF2 predictions in this study can be used.

Minor comments

• Line 283 – presumably this is 2.2Å?

Response: Thanks, we have made the correction (line 142) (Updated 142).

• The size exclusion chromatograms of refolded complexes clearly used different columns with different elution volumes - should note expected elution volume of refolded MHC-I for readers reference

Response: Thanks to your comment, we have added a note into the Figure legend (new Figure. 1a).

(vi) A number of high confidence peptides are reported containing Cysteine residues

(which are not supposed to be present in the random peptide libraries used to refold the MHC-I molecules) - if possible the authors should respond to this problem as well.

Response: Thank you for your careful review. This was also questioned by a Reviewer in a previous study. This problem is actually caused by De Novo MS result analysis, the non-data-dependent MS identification approach, just like mentioned in your question (ii).

In previous studies, we validated a number of peptides with high scores but clearly not conforming to the peptide binding motif, and none of them could bind. This suggested that the specific peptides obtained by De Novo MS sequencing should not be credible. When identifying specific peptide sequences, the De Novo algorithm has the possibility of misidentifying certain signals as cysteines. However, in terms of overall statistics, the results produced by the De Novo MS analysis are credible. There may be certain errors in a specific peptide in the random peptide library generated by RPLD-MS, but the overall characteristics of all random peptides do conform to the peptide binding motif of the tested MHC-I molecule, whether from the perspective of structure or in vitro refolding verification. From the overall statistics, cysteine appeared at a very low percentage and did not have a significant effect on the PBM logo.

Thanks again to the reviewers for their valuable suggestions on our article.

Reviewer #3 (Remarks to the Author):

In their manuscript, Wang and colleagues present evidence for structural changes on bat MHC molecules upon a 3- or 5-amino acid insertion. These insertions are proposed to stabilize the molecules, and they are part of the bat immune system.

In my review I will focus mainly on the structural work of this manuscript, because this is where my competences are.

The authors present several crystal structures. Unfortunately, they did not insert the main piece of evidence for the structural changes associated with the insertions. In my opinion it is imperative that the authors show an unbiased (2Fo-Fc)-electron density map of the region of interest. The authors should also comment on the B-factors of the atoms of this region. Are they higher, lower or at par with the B-factors of the other atoms of the structures. Only with this evidence it is possible to judge the claims of the authors.

Response: Thank you for your comments. We added the unbiased (2Fo-Fc)-electron density maps to the new figures 1, 2, and 4. We also added a B factor comparison in

the Results (new Figure. 4a and line 223) (Updated 227).

Minor comment here: the authors state an "impressive resolution of 2.2 Å". First of all, this is like a typo. 2.2 Å should be 2.2 Å. Second, the authors should refrain from using words such as "impressive". What is impressive for somebody, may not necessarily be impressive for the other. Considering that the average resolution of the structures in the PDB is about 2.0 Å, 2.2 Å does not sound as impressive anymore.

Response: Thanks, we have corrected (line 142) (Updated 142).

Anyways, the authors should also comment and explain the term B-factors in putty style, which is used in several figures. This is not familiar to everybody.

Response: We provided an introduction to the B-factor putty model (line 227) (Updated 231). In addition, as mentioned before, we also provided more comparisons of B factors in the region of interest (new Figure 4A and line 223) (Updated 227).

Finally, the authors should explain what kind of map corrections they did in Coot. In my view, Coot is a visualization program, it does not do any map corrections. Please be concise there.

Response: Coot is used to manually build the structural model based on the (2Fo-Fc)-electron density map. We have corrected, thanks for your comment.

Another aspect to criticize is the use of Alphafold2 predictions. These predictions should always be taken with a grain of salt. Of particular importance in this respect is the question, what is the confidence level in particular of the insertion part of the structures. Is this higher, lower or at par with the confidence level of the rest of the structure.

Response: Your comments are important. We added the presentation and description of the confidence of the predicted results (Figure. 6a and line 289). The confidence in the predicted Mylu-B*67:01 heavy chain structure predicted by AF2 is high (pLDDT>90), including the small helix formed by the MQQPW insertion. The overall confidence in the predicted complex structure is also high, with relatively low confidence in the binding peptide and the loop at ΔMQQPW (90 > pLDDT > 70). Only the exposing residues of Mylu-B*67:01ΔMQQPW bound peptide are with low confidence. These results suggested that AF2 was relatively accurate in its predictions of α-helices and β-sheets, while the confidence in its predictions of more flexible rings was relatively low.

Although the AF2 predictions cannot be equated to the resolved structures, we believe

that the predictions for bat MHC-I_s can be used in this study. In addition to the high confidence of the prediction results, sequence comparison revealed that these insertions are somewhat conserved, and the predicted results reflected this similarity (new Figure. 6c). In addition, there have been some studies applying AF2 to HLA structure prediction and epitope screening with good results (Discussion inside 44,45).

Since much of the manuscript is centered on the question of how the structural alterations bring about thermostability, these issues should be taken seriously by the authors. A figure there would certainly help.

Response: To make the logic clearer, we have modified the order in which the results are described. We first focused on elucidating the improved structural stability and then described its effect on pMHC-I thermal stability and peptide presentation. The old Figure. 5 and Figure. 6 have been newly recombined to become the new Figure 4 and Figure 5. This provides a clear illustration of how increased structural stability affects the thermostability of pMHC-I.

Overall, I find the manuscript a bit too lengthy for the content. Maybe the authors can think of moving some of the present figures to Supplementary Material. Electron density and AlphaFold confidence level should go to the main text.

Response: We reduced a figure in the main text based on your comments. Part of it was put into Supplementary Material, while the electron cloud density and AF2 confidence were put into the main text.

We sincerely thank the reviewers for their suggestions to us regarding the crystal structure.

We appreciate for Editors/Reviewers' warm work earnestly, and hope that the correction will meet with approval.

Once again, thank you very much for your comments and suggestions.

REVIEWERS' COMMENTS:

Reviewer #3 (Remarks to the Author):

The authors have now responded to all of my comments in a satisfactory manner and modified the manuscript accordingly. I have no further comments.